# FACTOR LEARNING PORTFOLIO OPTIMIZATION INFORMED BY CONTINUOUS-TIME FINANCE MODELS

## ABSTRACT

We study financial portfolio optimization in the presence of unknown and uncontrolled system variables referred to as stochastic factors. Existing work falls into two distinct categories: (i) reinforcement learning employs end-to-end policy learning with flexible factor representation, but does not precisely model the dynamics of asset prices or factors; (ii) continuous-time finance methods, in contrast, take advantage of explicitly modeled dynamics but pre-specify, rather than learn, factor representation. We propose FaLPO (factor learning portfolio optimization), a framework that interpolates between these two approaches. Specifically, FaLPO hinges on deep policy gradient to learn a performant investment policy that takes advantage of flexible representation for stochastic factors. Meanwhile, FaLPO also incorporates continuous-time finance models when modeling the dynamics. It uses the optimal policy functional form derived from such models and optimizes an objective that combines policy learning and model calibration. We prove the convergence of FaLPO and provide performance guarantees via a finite-sample bound. On both synthetic and real-world portfolio optimization tasks, we observe that FaLPO outperforms five leading methods. Finally, we show that FaLPO can be extended to other decision-making problems with stochastic factors.

## 1 INTRODUCTION

Portfolio optimization studies how to allocate investments across multiple risky financial assets such as stocks and safe assets such as US government bonds. The investment target is often formulated as maximizing the expected *utility* of the investment portfolio's value at a fixed time horizon, which conceptually maximizes profit while constraining risk (von Neumann & Morgenstern, 1947). With continuous-time stochastic models of stock prices, great advances in the expected utility maximization framework were made in Merton (1969) using stochastic optimal control (dynamic programming) methods. More realistic models incorporate *factors* like economic indices and proprietary trading signals (Merton et al., 1973; Fama & French, 2015; 1992), which (i) affect the dynamics of stock prices; (ii) stochastically evolve over time; (iii) are not affected by individual investment decisions. With greater data availability, it is natural to design and apply data-driven machine learning methods (Bengio, 1997; Dixon et al., 2020; De Prado, 2018) to handle factors for portfolio optimization. This work proposes a novel method—Factor Learning Portfolio Optimization (FaLPO)—which combines tools from both machine learning and continuous-time finance.

Portfolio optimization with stochastic factors is challenging for three reasons. First, financial data is notoriously noisy and idiosyncratic (Goyal & Santa-Clara, 2003), causing complex purely data-driven methods to be unstable and prone to overfitting. Second, the relationship between the factors and their impact on stock prices can be extremely complicated and difficult to model *ex ante*. Third, many successful finance models are in continuous time and require interacting with the environment infinitely frequently. As a result, such models cannot be easily combined with machine learning methods, many of which are in discrete time.

Current approaches to portfolio optimization broadly fall into two categories: reinforcement learning (RL) and continuous-time finance methods. Many RL solutions to portfolio optimization are built on deep deterministic policy gradient (Silver et al. 2014; Hambly et al. 2021, Section 4.3). Such methods parameterize the policy function as a neural network with strong representation power and learn the neural network by optimizing the corresponding portfolio performance. However, these approaches

(as well as other model-free methods like Haarnoja et al. 2018) have high sample complexity and tend to overfit due to the high noise in the data. Other RL methods explicitly learn representation (Watter et al., 2015; Lee et al., 2020; Laskin et al., 2021) and leverage discrete-time models (Deisenroth & Rasmussen, 2011; Gu et al., 2016; Mhammedi et al., 2020; Janner et al., 2019; Nagabandi et al., 2018). Nonetheless, these methods are not informed by continuous-time finance models and, as our experiments suggest in Section 5, cannot benefit from structures inherent in the financial market.

Stochastic factor models can be used to mathematically derive optimal (or approximately optimal) investment policies (Kim & Omberg, 1996; Chacko & Viceira, 2005; Fouque et al., 2017; Avanesyan, 2021). To this end, one needs domain knowledge to pick and model the factors. Then, *model calibration* (a.k.a. model fitting, parameter estimation) is conducted by maximizing a calibration objective. With the calibrated model, the optimal investment policy can be derived analytically or numerically (Merton, 1992; Fleming & Soner, 2006). This procedure of calibration and optimization effectively constrains the 'learning' in the optimization step, and thus helps reduce overfitting to noisy data. However, this approach cannot capture the complicated factor effects in the data, because the factors may be complex and unlikely to be identified manually. Therefore, these methods may end up with oversimplified models and suffer from model bias with suboptimal performance.

To tackle these limitations, we propose factor learning portfolio optimization (FaLPO), a new method that interpolates between the two aforementioned solutions (Figure 1). FaLPO includes (i) a neural stochastic factor model to handle huge noise and complicated factor effects and (ii) a model-regularized policy learning method to combine continuous-time models with discrete-time policy learning methods. First, to reduce the sample complexity and avoid overfitting, FaLPO assumes factors and asset prices follow a parametric continuous-time finance model. To capture the complicated factor effects, FaLPO models

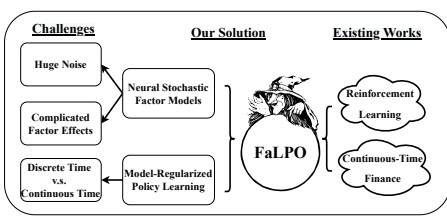

Figure 1: Demonstration of FaLPO

the factors by a representation function $\phi$ parameterized by a neural network with minimal parametric constraints. Second, for policy learning, FaLPO incorporates two regularizations derived from continuous-time stochastic factor models: a policy functional form and model calibration. Specifically, we derive policy functional forms from the neural stochastic factor model using stochastic optimal control tools, and apply it to parameterize the candidate policy in FaLPO. The use of this form in the learning algorithm effectively acts as a regularizer. Then, model calibration and policy learning are conducted jointly, such that the learned policy is informed by continuous-time models.

Theoretically, we prove that the added continuous-time regularization leads to the optimal portfolio performance as the trading frequency increases. Empirically, we demonstrate the improved performance of the proposed method by both synthetic and real-world experiments. We review the related literature in Appendix A. We also discuss how FaLPO is extendable beyond portfolio optimization, and can be applied to other decision-making problems with stochastic factors in Appendix H.

## 2 BACKGROUND

In this section, we first formulate the portfolio optimization problem. We then review two major solutions to this problem: deep deterministic policy gradient in reinforcement learning (RL) and stochastic factor models in continuous-time finance.

### 2.1 PORTFOLIO OPTIMIZATION

**Problem Formulation**    Portfolio optimization seeks to derive a policy of asset allocation that yields high return while maintaining low risk for the investment. Formally, consider $d_S$ risky assets with prices $S_t := [S_t^1, S_t^2, \cdots S_t^{d_S}]^\top$ and a risk-free money market account with, for simplicity, zero interest rate of return (like cash). We observe $d_Y$ features (e.g. economic indices, market benchmarks) denoted as $Y_t$. From $Y_t$, we can derive $d_X$ factors denoted as $X_t$ which (i) affect the dynamics of asset prices; (ii) evolve over time stochastically; (iii) are not affected by investment decisions. Given an initial investment capital (or wealth) $z_0$ and the initial values for $Y_t$ and $S_t$ as $y_0$ and $s_0$, we use a

$d_S \times 1$ vector $\pi_t$ to denote the fractions of wealth invested in the $d_S$ assets at time point $t$. Note that negative values are allowed in $\pi_t$ indicating short positions. At the terminal time $T > 0$, the target is to maximize the expectation of a given utility function $\mathbb{E}[U(Z_T^\pi)]$, where $U : \mathcal{Z} \to \mathbb{R}$ with $\mathcal{Z} \subseteq \mathbb{R}$ is the utility function and $Z_T^\pi$ denotes the terminal wealth under $\pi$.

Intuitively, a utility function reflects the risk preference of an investor. It is an increasing function of wealth that is also concave: it changes significantly when the wealth is low but less so when the wealth is high (Figure 2). This work focuses on the power utility $U(z; \gamma) := \frac{1}{1-\gamma} z^{1-\gamma}$ with $\mathcal{Z} = \mathbb{R}^+$, $\gamma > 0$, and $\gamma \neq 1$; and the exponential utility $U(z; \gamma) := -\frac{\exp(-\gamma z)}{\gamma}$ with $\mathcal{Z} = \mathbb{R}$ and $\gamma > 0$. Here, $\gamma$ is the investor's risk aversion coefficient and is hand-picked (instead of tuned) by the user. A larger $\gamma$ corresponds to more risk aversion, while a smaller $\gamma$ cor-

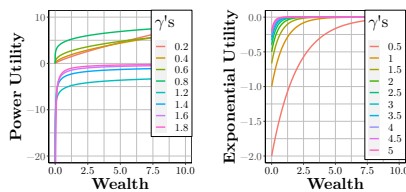

Figure 2: Power & exponential utilities.

responds to more risk tolerance. Beyond these two utilities, our method is also applicable to other utility functions and other objective functions for portfolio optimization (see Appendix B).

**Discrete- and Continuous-Time Policies**  Discrete- and continuous-time policies are two major types of investment policies, differing on how frequently the portfolio is rebalanced. A discrete-time policy rebalances the portfolio fintely frequently, leading to a discrete-time dynamics for the wealth. Such policies are often considered in RL methods (Section 2.2). Continuous-time policies rebalance the investment infinitely frequently, leading to a continuous-time dynamics for the wealth. These policies are often found explicitly in continuous-time finance models (Section 2.3) [1].

## 2.2 DEEP DETERMINISTIC POLICY GRADIENT

We review deep deterministic policy gradient (DDPG, Silver et al. 2014)—a quintessential example of RL methods. DDPG does not explicitly model the dynamics, but instead directly learns a discrete-time policy for portfolio optimization. DDPG parameterizes the policy function as a deep neural network and conducts gradient-based policy learning. Denote by $\pi(t, S_t, Z_t, Y_t; \theta_D)$ the deep policy function with parameter $\theta_D$. Without explicitly modeling the dynamics of $S_t$ or $X_t$, DDPG directly maximizes the following performance objective to learn a policy:

$$\max_{\theta_D} V(\theta_D) \text{ with } V(\theta_D) := \mathbb{E}[U(Z_T^{\pi(\cdot;\theta_D)})], \tag{1}$$

where the expectation is over the terminal wealth $Z_T^{\pi(\cdot;\theta_D)}$ following the policy $\pi(\cdot; \theta_D)$. A key step of DDPG is to compute the gradient of $V(\theta_D)$ to update $\theta_D$. Following the procedure in Appendix C, this can be achieved by sampling the trajectories of $S_t$ and $Y_t$ to approximate the expectation and thus the gradient of $V(\theta_D)$.

Typically, DDPG learns a discrete-time policy that rebalances the portfolio finitely frequently. To see how the policy rebalances the portfolio, we study its corresponding wealth process $Z_t^{\pi(\cdot;\theta_D)}$ that characterizes the changes in wealth over time. Let $\Delta t > 0$ be the time interval (e.g. daily, weekly) to rebalance the portfolio and, for integer $M > 0$, let $T := M\Delta t$ be the fixed investment horizon (e.g. one or two months). At time $m\Delta t$ with $m \in \{0, 1, 2, \cdots, M-1\}$, define $\pi_{m\Delta t}^i := \pi^i(m\Delta t, S_{m\Delta t}, Z_{m\Delta t}, Y_{m\Delta t}; \theta_D)$ as the fraction of current wealth invested in the $i^{th}$ risky asset. Then, the wealth change at time $m\Delta t$ is: $Z_{(m+1)\Delta t}^{\pi(\cdot;\theta_D)} - Z_{m\Delta t}^{\pi(\cdot;\theta_D)} = Z_{m\Delta t}^{\pi(\cdot;\theta_D)} \left[ \sum_{i=1}^{d_S} \pi_{m\Delta t}^i \frac{S_{(m+1)\Delta t}^i - S_{m\Delta t}^i}{S_{m\Delta t}^i} \right]$, where $Z_{m\Delta t}^{\pi(\cdot;\theta_D)} \pi_{m\Delta t}^i \frac{S_{(m+1)\Delta t}^i - S_{m\Delta t}^i}{S_{m\Delta t}^i}$ is the wealth change due to the investment in the $i^{th}$ risky asset. Note that the number of shares invested in an asset $(Z_{m\Delta t}^{\pi(\cdot;\theta_D)} \frac{\pi_{m\Delta t}^i}{S_{m\Delta t}^i})$ does not change during $(m\Delta t, (m+1)\Delta t)$: the portfolio rebalances every $\Delta t$ time.

RL methods like DDPG provide flexible representation for factors: the hidden layers of the neural network are considered as the representation learned for $Y_t$, providing strong representation power.

---

[1]Note that it is impossible to rebalance a portfolio infinitely frequently in practice. Thus, continuous-time policies are more useful as analytical tools.

Nonetheless, there is not an explicit parametric model for the learned representation and asset prices. Consequently, such methods require lots of data and tend to overfit (Aboussalah, 2020).

## 2.3 Stochastic Factor Models

We review stochastic factor models in continuous-time finance. These models can explicitly formulate the dynamics and can also be used to mathematically derive the functional form of the optimal continuous-time investment policy. Stochastic factor models are described by stochastic differential equations (SDEs) (see Oksendal 2013 and Appendix D) to formulate the dynamics of asset prices $S_t$. Specifically, with a $d_X \times 1$ factor variable $X_t$, let $W_t := [W_t^1, W_t^2, \cdots W_t^{d_W}]^\top$ be a $d_W \times 1$ Brownian motion that characterizes random fluctuations. Then, $S_t$ and $X_t$ are assumed to follow

$$
\frac{dS_t^i}{S_t^i} = f_S^i(X_t; \theta_S^*)dt + \sum_{j=1}^{d_W} g_S^{ij}(X_t; \theta_S^*)dW_t^j, \quad i \in \{1, 2, \cdots, d_S\},
$$
$$
dX_t = f_X(X_t; \theta_S^*)dt + g_X(X_t; \theta_S^*)^\top dW_t.
$$
(2)

In (2), $f_S : \mathbb{R}^{d_x} \to \mathbb{R}^{d_S}$, $f_X : \mathbb{R}^{d_x} \to \mathbb{R}^{d_x}$, $g_S : \mathbb{R}^{d_x} \to \mathbb{R}^{d_S \times d_W}$, and $g_X : \mathbb{R}^{d_x} \to \mathbb{R}^{d_x \times d_W}$ are parametric functions pre-specified by domain knowledge. Further, $f_S$ and $f_X$ are often referred to as the drift, and $g_S$ and $g_X$ as the volatility of $S_t$ and $X_t$ respectively. Intuitively, SDEs formulate the change of a variable in an infinitesimal time step as the sum of a deterministic part ($dt$) and a stochastic part ($dW_t$), and we use $\theta_S^*$ to parameterize the SDE. The factor $X_t$ appears in both the drift and volatility of the asset prices, thus affecting the price transition. With the parametric functional forms in (2), we can use tools in stochastic optimal control to derive the functional form of the optimal continuous-time investment policy.

Continuous-time policies change the investment in each asset at every time point. For a continuous-time investment policy $\tilde{\pi}_t$, the dynamics of wealth $\tilde{Z}_t^{\tilde{\pi}}$ is defined as $\frac{d\tilde{Z}_t^{\tilde{\pi}}}{\tilde{Z}_t^{\tilde{\pi}}} := \sum_{i=1}^{d_S} \frac{\tilde{\pi}_t^i dS_t^i}{S_t^i}$, with $\tilde{Z}_0 = z_0$, $S_0 = s_0$ and $X_0 = x_0$. Crucially, this is different from the discrete-time wealth process $Z_t^\pi$ in Section 2.2, as the number of shares $\frac{\tilde{Z}_t^{\tilde{\pi}} \tilde{\pi}_t^i}{S_t^i}$ in asset $i$ now changes continuously over time, as opposed to being rebalanced at finite intervals. This discrepancy creates obstacles to directly apply the results derived from stochastic factor models to discrete-time policy learning.

Stochastic factor models can reduce sample complexity for portfolio optimization, since the assumed functional forms in (2) significantly constrain the solution space. However, a crucial step to apply stochastic factor models is to pick or even construct $X_t$ from the observed $Y_t$ that perfectly follows a pre-specified model. This step often relies on domain knowledge and thus may end up with oversimplified models suffering from model bias and eventually leading to suboptimal performance.

## 3 Factor Learning Portfolio Optimization

We propose factor learning portfolio optimization (FaLPO), a new decision-making framework that interpolates between DDPG and stochastic factor models. FaLPO has two components: (i) a neural stochastic factor model to handle huge noise and complicated factor effects and (ii) model-regularized policy learning to combine continuous-time models with discrete-time policy learning methods.

## 3.1 Neural Stochastic Factor Models

We describe neural stochastic factor models (NSFM) and discuss their benefits. On the one hand, a neural stochastic factor model assumes the existence of a representation function $\phi$ such that the factors of the problem can be directly learned from its features: $X_t = \phi(Y_t; \theta_\phi^*)$. Here, $\phi$ is formulated as a neural network with parameter $\theta_\phi^*$ (Figure 3). As a result, FaLPO avoids hand-picking factors from features as is the case in stochastic factor models (Section 2.3). The neural network representation has only a few parametric constraints and thus is able to capture complicated factor effects in the data. Furthermore, factors $X_t$ and asset prices $S_t$ are assumed to follow a stochastic factor model (e.g. (2) and (6)), which reduces the sample complexity and avoids overfitting.

## 3.2 MODEL-REGULARIZED POLICY LEARNING

Under the proposed neural stochastic factor model, we aim to learn the discrete-time optimal policy function $\pi_t^*$ and the representation function $\phi(\cdot; \theta_\phi^*)$. However, while the policy learning is for discrete-time policies, our proposed model is in continuous time. To bridge this gap, we incorporate two types of continuous-time model regularization into discrete-time policy learning: (i) the policy functional form (3) and (ii) the model calibration objective (4).

**Policy Functional Form** From our model, we apply the functional form of a continuous-time optimal policy into our discrete-time policy learning. Using tools in stochastic optimal control, we can derive the functional form of an optimal continuous-time policy: $\tilde{\pi}_t^* = \Pi(t, S_t, Z_t, X_t; \theta_{\tilde{\pi}}^*)$, where the functional form of $\Pi$ can be obtained in many existing stochastic factor models (Kim & Omberg, 1996; Chacko & Viceira, 2005; Avanesyan, 2021; Zariphopoulou, 2001; Wachter, 2002; Kraft, 2005), and $\theta_{\tilde{\pi}}^*$ is an optimal parameter for $\Pi$. FaLPO uses the functional form of $\Pi$ in policy learning and parameterize the candidate policy as

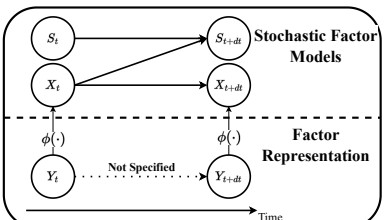

Figure 3: Demonstration for neural stochastic factor models.

$$\pi(t, S_t, Z_t, Y_t; \theta_\phi, \theta_\pi) := \Pi(t, S_t, Z_t, \phi(Y_t; \theta_\phi); \theta_\pi), \tag{3}$$

where $\phi$ is the representation function for the factors in Section 3.1. As a result, $\Pi$ constrains the policy space and acts as regularization. Importantly, although $\Pi$ is derived for continuous-time policies, it can still provide guidance for discrete-time policy learning when $\Delta t$ is small. We rigorously prove the soundness of using (3) in Section 4.

**Model Calibration** FaLPO also hinges on model calibration to regularize policy learning. Given the specific functional forms in (2), FaLPO conducts model calibration to estimate the parameters of the SDE. The calibration procedure can be summarized as maximizing a model calibration objective:

---

**Algorithm 1** FaLPO

1: **Input:** number of iterations $N$.
2: Initialize $\theta_\phi$ and $\theta_\pi$ .
3: **for** $n \in [N]$ **do**
4:     Parameterize the policy function according to (3).
5:     Estimate the policy gradient for $H$ in (5) (Appendix C).
6:     Update $\theta_\phi$, $\theta_\pi$, and $\theta_S$.
7: **end for**
8: **Return** $\pi(\cdot; \boldsymbol{\theta}_\phi, \boldsymbol{\theta}_\pi)$

---

$$\max_{\theta_S} L(\theta_\phi, \theta_S). \tag{4}$$

In practice, with discrete data, one may use likelihood (Phillips, 1972; Beskos & Roberts, 2005) or other likelihood-based objective functions (Bishwal, 2007; Ait-Sahalia & Kimmel, 2010) for $L$ (see Appendix E for concrete examples).

To harness the information provided by model calibration in policy learning, FaLPO combines the model calibration objective $L$ in (4) with the performance objective $V$ in (1) and facilitates a joint optimization over the two. Note that naively combining $L(\theta_\phi, \theta_S)$ and $V(\theta_D)$ is not effective since the two in general do not share common parameters: the parameter of the policy network $\theta_D$ has no overlap with the SDE parameter $\theta_S$ or factor representation parameter $\theta_\phi$. However, by constraining the policy space to (3) in FaLPO, we can show that $\theta_\phi$ is also part of the policy parameterization. Thus, $V$ can be derived as $V(\theta_\phi, \theta_\pi) := \mathbb{E}[U(Z_T^{\pi(\cdot; \theta_\phi, \theta_\pi)})]$. In other words, $\theta_\phi$ is shared in both $V$ and $L$, and hence FaLPO can carry out a joint optimization over the two:

$$\max_{(\theta_\phi, \theta_\pi, \theta_S) \in \mathcal{A}} H(\theta_\phi, \theta_\pi, \theta_S), \text{ with } H(\theta_\phi, \theta_\pi, \theta_S) := (1 - \lambda) V(\theta_\phi, \theta_\pi) + \lambda L(\theta_\phi, \theta_S), \tag{5}$$

where the candidate policy follows the functional form of the optimal continuous-time policy (3) (see Algorithm 1), and $\mathcal{A}$ denotes the considered parameter set. The model calibration objective also acts as a model regularization, where $\lambda \in (0, 1)$ is a hyperparameter determining its effect. In practice, we can optimize (5) by gradient-based methods, facilitating a easy and end-to-end learning procedure (see Appendix C for gradient estimation details).

### 3.3 EXAMPLE OF FALPO

In portfolio optimization, one can use different types of stochastic factor models. FaLPO can be applied to many such types (Appendix F). In this section, we use the Kim–Omberg model (Kim & Omberg, 1996) as an example to illustrate FaLPO's modeling and policy learning. Kim–Omberg is a standard model for portfolio optimization with stocahstic factors, which has been extensively studied empirically (Welch & Goyal, 2008; Muhle-Karbe et al., 2017). For modeling, FaLPO with Kim–Omberg model formulates the dynamics of asset prices and factors as

$$\frac{dS_t^i}{S_t^i} = X_t^i \, dt + \sum_{j=1}^{d_W} \sigma^{ij} \, dW_t^j, \, dX_t = \mu(\omega - X_t) \, dt + v \, dW_t, \text{ and } X_t = \phi(Y_t; \theta_\phi^*), \quad (6)$$

where SDE parameters $\omega$, $\sigma$, $v$, and $\mu$ are constant matrices or vectors.

For policy learning, we detail the policy functional form and model calibration. Under the Kim-Omberg model, we can derive the optimal policy functional form $\Pi$ in (3). Specifically, for power utility $\Pi(t, S_t, Z_t, \phi(Y_t; \theta_\phi); \theta_\pi) = k_1(t; \theta_\pi)\phi(Y_t; \theta_\phi) + k_2(t; \theta_\pi)$, for exponential utility $\Pi(t, S_t, Z_t, \phi(Y_t; \theta_\phi); \theta_\pi) = k_1(t; \theta_\pi)\phi(Y_t; \theta_\phi)/Z_t + k_2(t; \theta_\pi)/Z_t$, where $k_1(\cdot; \theta_\pi) : [0, T] \to \mathbb{R}^{d_S \times d_X}$ and $k_2(\cdot; \theta_\pi) : [0, T] \to \mathbb{R}^{d_S \times d_X}$ are two time dependent functions (Appendix F.2). We can derive the functional forms of $k_1$ and $k_2$ since the two are solutions to systems of ODEs related to algebraic Riccati equations (Appendix G). We can also directly use function approximators like neural networks or kernel methods for the two. For model calibration, we use a negative mean square loss with the derivation deferred to Appendix E: $L(\theta_\phi, \theta_S) :=$
$-\mathbb{E}\left[ \sum_{i=1}^{d_S} \left[ \log(S_{t+\Delta t}^i) - \log(S_t^i) - \phi^i(Y_t; \theta_\phi)\Delta t - \theta_S^i \right]^2 \right]$, where in this case $\theta_S$ is a $d_S \times 1$ vector. Therefore, to implement FaLPO, we can parameterize the candidate policy function using $\Pi$ and optimize (5).

Note that the methodology of FaLPO is also generally applicable to other decision-making problems besides portfolio optimization. In Appendix H, we use linear quadratic control with stochastic factors as an example to demonstrate the generality of FaLPO.

## 4 THEORY

We theoretically analyze both the asymptotic and non-asymptotic characteristics of FaLPO.

### 4.1 ASYMPTOTIC ANALYSIS

FaLPO applies the policy functional form and model calibration derived from continuous-time models to discrete-time policy learning. We show that FaLPO can achieve the optimal performance asymptotically (i.e. with infinite data and perfect optimization). In the following, we describe the assumptions and results and provide the formal theorem in the end.

We provide an intuitive description on the assumptions, with the formal statements provided in Appendix I.1. First, we assume that the portfolio optimization problem satisfies some standard regularity conditions (Higham et al., 2002): the drift and volatility are locally Lipschitz continuous; meanwhile, the asset prices, the stochastic factors, and the wealth process under the optimal policy have bounded moments. Second, we assume that the utility function $U(z)$ has linear growth on $z \in \mathcal{Z}$. Note that some widely used cases like power utility with $\gamma < 1$ and exponential utility with lower-bounded wealth satisfy this assumption. Third, we consider only admissible policies with parameters in $\mathcal{A}$ and we assume that $\mathcal{A}$ covers the optimal continuous-time policy parameter. A policy is admissible if it is predictable and if the wealth process $Z_t^\pi \in \mathcal{Z}$ for any $t \in [0, T]$ almost surely. It is a common practice to only consider such admissible policies in portfolio optimization. The last two assumptions are artifacts of the current theoretical analysis; in practice FaLPO can achieve reasonable performance without enforcing them (see Section 5).

With the foregoing assumptions, we show that the performance of FaLPO can asymptotically converge to that of the best policy in discrete time. In detail, we define $V_{\Delta t}^* := V(\pi^*)$ where $\pi^*$ is an optimal discrete-time admissible policy with time interval $\Delta t$, i.e., $V_{\Delta t}^*$ is the optimal performance obtained without constraining to the functional form (3) or leveraging model calibration like (5). Next, define

$\theta^*_{\Delta t} := (\theta^*_{\phi, \Delta t}, \theta^*_{\pi, \Delta t}, \theta^*_{S, \Delta t}) \in \arg \max_{(\theta_\phi, \theta_\pi, \theta_S) \in \mathcal{A}} H(\theta_\phi, \theta_\pi, \theta_S)$ with the policy functional form (3), such that $V(\theta^*_{\Delta t})$ is the performance that FaLPO can achieve with infinite data and perfect optimization. Then, Theorem 4.1 shows that the gap between $V^*_{\Delta t}$ and $V(\theta^*_{\Delta t})$ converges to zero as $\Delta t$ goes to zero.

THEOREM 4.1  With assumptions in Appendix I.1, $\lim_{\Delta t \to 0} \left( V^*_{\Delta t} - V(\theta^*_{\Delta t}) \right) = 0$.

Theorem 4.1 justifies the methodology of FaLPO under a small time interval. The proof is provided in Appendix I.2 and Appendix I.3.

## 4.2  NON-ASYMPTOTIC ANALYSIS

We study the finite-sample performance of FaLPO. We describe the problem setup, major assumptions, and then provide the theorem. In each iteration, we collect $B$ independent trajectories to estimate the gradient of $H$. Let $\theta_n$ be the estimate after the $n^{\text{th}}$ iteration, and $N$ the total number of iterations. we analyze the average estimate $\bar{\theta} := \frac{\sum_{n=1}^N \theta_n}{N}$ instead of $\theta_N$, which is a common technique for stochastic optimization analysis. Specifically, we aim to bound the expected difference between $V^*_{\Delta t}$ and $V(\bar{\theta})$. Note that it is extremely challenging to theoretically analyze a non-convex stochastic optimization (5) without further specifications in problem setup and assumptions (Polyak, 1963; Bhandari & Russo, 2019; Jin et al., 2021; Ma, 2020; Wang et al., 2019). Therefore, we consider a projection-based variant of FaLPO, under which the optimization process is conducted in a bounded parameter set $\mathcal{B} \subseteq \mathcal{A}$. Furthermore, we assume that the objective function $H$ is strongly concave in $\mathcal{B}$ with a local maximal point $\theta^\dagger_{\Delta t} := (\theta^\dagger_{\phi, \Delta t}, \theta^\dagger_{P, \Delta t}, \theta^\dagger_{S, \Delta t})$. Similar local curvature assumptions are commonly used to analyze non-convex problems (Bach et al., 2017; Loh, 2017). With the above setup and assumptions, the expected gap between $V^*_{\Delta t}$ and $V(\bar{\theta})$ satisfies the following finite-sample bound in Theorem 4.2.

THEOREM 4.2  With the aforementioned projection-based FaLPO algorithm and assumptions, both detailed in Appendix J.2, there exist positive constants $C_1$, $C_2$, $C_3$, and $C_4$ such that

$$\mathbb{E}[V^*_{\Delta t} - V(\bar{\theta})] \leq \frac{e_{\Delta t}}{1 - \lambda} + \frac{H(\theta^*_{\Delta t}) - H(\theta^\dagger)}{1 - \lambda} + \frac{C_1 \log(N)}{N(1 - \lambda)}$$
$$+ \frac{C_1 \log(N)}{BN(1 - \lambda)} \left[ (1 - \lambda)^2 C_2 + \lambda^2 C_3 + 2\lambda(1 - \lambda) C_4 \right], \tag{7}$$

where $\lambda \in [0, 1]$. Also, $e_{\Delta t}$ is an error term not related to $N$ or $B$ but dependent on $\Delta t$ with $\lim_{\Delta t \to 0} e_{\Delta t} = 0$.

Theorem 4.2 provides a non-asymptotic upperbound on the gap between the optimal performance $V^*_{\Delta t}$ and the one achieved by FaLPO $V(\bar{\theta})$. We briefly comment on each term in the upperbound. $\frac{e_{\Delta t}}{1-\lambda}$ bounds the asymptotic performance gap caused by leveraging the continuous-time policy functional form constraint and model calibration, and we explain its connection to Theorem 4.1 in Appendix J.4. $\frac{H(\theta^*_{\Delta t}) - H(\theta^\dagger_{\Delta t})}{1-\lambda}$ controls the performance gap between the local optimal point $\theta^\dagger_{\Delta t}$ and the global optimal point $\theta^*_{\Delta t}$. The remaining terms characterize the performance gap between $\bar{\theta}$ and $\theta^\dagger_{\Delta t}$.

Theorem 4.2 has two implications. First, the bound in (7) is a rational function of $\lambda$. Accordingly, there exist situations where a $\lambda \in (0, 1)$ provides a smaller upper bound than $\lambda = 0$, indicating the possibility that tuning $\lambda$ can provide better performance (see experiments in Appendix J). Second, when $\lambda = 1$, the bound diverges to infinity. This makes sense since when $\lambda = 1$, $H(\theta_\phi, \theta_\pi, \theta_S) = L(\theta_\phi, \theta_S)$ does not contain $\theta_\pi$: the algorithm does not learn the policy. We prove the theorem in Appendices J.1, J.3 and J.4.

## 5  EXPERIMENTS

By incorporating continuous-time finance models into policy learning, FaLPO can deal with high data noise and complex factor effects. In this section, we demonstrate the improved performance of FaLPO against existing portfolio optimization methods, over synthetic and real-world experiments.

| Methods | Explicit Factor Representation | Continuous-Time Model | Discrete-Time Model |
|---------|-------------------------------|----------------------|---------------------|
| MMMC | ✘ | ✔ | ✘ |
| DDPG | ✘ | ✘ | ✘ |
| SLAC | ✔ | ✘ | ✔ |
| RichID | ✔ | ✔ | ✘ |
| CT-MB-RL | ✘ | ✔ | ✘ |
| **FaLPO** | ✔ | ✔ | ✘ |

Table 1: Competing methods and their characteristics.

| Annual Volatility | 0.1 | 0.2 | 0.3 |
|-------------------|-----|-----|-----|
| **FaLPO** | $\mathbf{-0.465} \pm 0.446$ | $\mathbf{-1.35} \pm 0.155$ | $\mathbf{-2.737} \pm 0.219$ |
| DDPG | $-1.650 \pm 0.456$ | $-3.30 \pm 1.294$ | $-5.495 \pm 1.269$ |
| SLAC | $-0.750 \pm 0.210$ | $-5.50 \pm 0.011$ | $-6.160 \pm 0.012$ |
| RichID | $-3.350 \pm 0.111$ | $-5.65 \pm 0.102$ | $-6.325 \pm 0.048$ |
| CT-MB-RL | $-2.850 \pm 0.014$ | $-5.35 \pm 0.020$ | $-6.160 \pm 0.026$ |
| MMMC | $-4.723 \pm 7.619$ | $-5.602 \pm 4.299$ | $-6.124 \pm 3.217$ |

Table 2: Average terminal utility after tuning with standard deviation for synthetic data

## 5.1 SYNTHETIC EXPERIMENTS

**Metrics** We compare different methods using the average terminal utility since it is the ultimate goal in our portfolio optimization problem formulation and is commonly used in continuous-time finance models. There exist other statistics measuring the performance of portfolios (see Section B). These statistics are not equivalent or consistent with the utility, and thus we do not emphasize them.

**Methods** We compare FaLPO with five competing methods representing various approaches in prior work (Table 1). (i) *Merton Model with Model Calibration (MMMC)*: model calibration of a classic continuous-time finance baseline, which does not consider stochastic factors (Merton, 1969), combined with closed-from policy function. (ii) *Deep Deterministic Policy Gradient (DDPG)*: a state-of-the-art model-free RL method with deterministic policy (Silver et al., 2014), which many empirical portfolio optimization methods build on. (iii) *Stochastic latent actor critic (SLAC)*: a state-of-the-art representation learning RL method that explicitly learns representation of latent variables (Lee et al., 2020). (iv) *Model-based RL with rich observations (RichID)*: a state-of-the-art model-based RL method with representation learning (Mhammedi et al., 2020). (v) *Continuous-time model-based RL (CT-MB-RL)*: policy gradient optimizing the performance objective using the policy functional form derived from continuous-time models, but directly treating the features as the factors.

For policy gradient methods, we pick a deterministic policy approach like DDPG as, when compared to non-deterministic policy gradient alternatives, they are more suitable to portfolio optimization due to the continuous action space, high exploration cost, and high noise in financial data (Appendix A.2). For portfolio optimization, different variance reduction methods for policy gradient (Schulman et al., 2015; 2017; Xu et al., 2020) only provide minor performance improvements (Aboussalah, 2020). We hence do not report such results. For areas of RL with representation learning and model-based RL, we focus on those (SLAC and RichID) explicitly learning a representation of latent variables, since such methods are more closely related to FaLPO. There are other techniques like data augmentation, feature construction, adversarial training, and regularization that can improve the empirical performance of portfolio optimization (see survey in Hambly et al., 2021, Section 4.3). In this work, we focus on the central methodological task of policy learning, and most such techniques can be directly combined with our proposed method.

**Protocol** We simulate environments with the Kim–Omberg model and implement the considered methods to compare their performance. Note that a key data generating parameter in portfolio optimization is the signal-to-noise ratio, which can be roughly characterized by the ratio between the scale of the drift and the scale of the volatility (see Appendix K.1 for detailed explanation). We test our method under different signal-to-noise ratios. To this end, we randomly generate stock drifts to around $10\%$ (to mimic the real-world average return of stocks in SP500), vary the scale of volatility in $\{10\%, 20\%, 30\%\}$, and simulate data following the procedure in Appendix K.2. Then, we apply the considered methods to maximize the terminal power and exponential utility with different $\gamma$'s. For each method, we tune the learning rate and other method-specific hyperparameters with early stopping (Appendix K.3). With each method-hyperparameter-environment combination, we repeat training, validation, and testing five times.

**Results** For exponential utility maximization, Table 2 summarizes the average test utility after hyperparameter selection with 10 stocks, 10 features, and $\gamma = 5$. FaLPO outperforms all the competing methods in terms of average terminal utility. This performance gain may be explained by the factor representation learning informed by the continuous-time finance model, as other methods are incapable of doing so. Meanwhile, MMMC and CT-MB-RL underperform, which suggests the disadvantage of using oversimplified models. Compared with the more sophisticated RL methods like SLAC and RichID, the simple DDPG is fairly competitive. This is consistent

| Methods | Energy | Material | Industrials | Mix |
|---------|--------|----------|-------------|-----|
| **FaLPO** | $-\mathbf{2.4} \pm 1.9$ | $-\mathbf{3.2} \pm 1.0$ | $-\mathbf{6.3} \pm 2.3$ | $-\mathbf{3.5} \pm 1.5$ |
| DDPG | $-6.6 \pm 1.2$ | $-7.3 \pm 1.5$ | $-7.3 \pm 2.1$ | $-2.5 \times 10^4 \pm 3.3 \times 10^8$ |
| SLAC | $-6.8 \pm 0.2$ | $-7.0 \pm 1.5$ | $-342.4 \pm 886.8$ | $-3.0 \times 10^8 \pm 4.3 \times 10^{12}$ |
| RichID | $-6.5 \pm 0.1$ | $-6.9 \pm 1.4$ | $-6.9 \pm 0.4$ | $-8.1 \pm 3.9$ |
| CT-MB-RL | $-4.2 \pm 6.2$ | $-5.4 \pm 4.3$ | $-11655 \pm 32947.5$ | $-5.7 \pm 3.1$ |
| MMMC | $-8.5 \pm 7.6$ | $-6.5 \pm 1.7$ | $-11.0 \pm 5.4$ | $-7.5 \pm 4.4$ |

Table 3: Average terminal utility for real-world data. Mix denotes a mix of stocks in the previous three sectors.

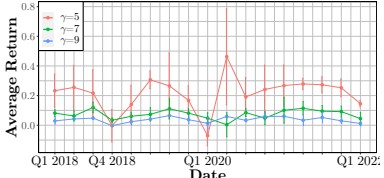

Figure 4: FaLPO average return over portfolio terminal dates.

with the existing observation that more complicated RL solutions may not always be suitable for portfolio optimization due to the large noise and idiosyncrasy in the data. In the appendix, we provide additional experimental results with different problem dimensions (Appendix K.4.1), other $\gamma$ values (Appendix K.4.2), and different utility functions (Appendix K.4.3). The results are consistent. Appendix K.4.4 studies a simplified case, where the optimal performance can be mathematically derived. FaLPO achieves a similar performance as the theoretically optimal one.

## 5.2 REAL-WORLD STOCK TRADING

In this section, we present an application of FaLPO for real-world stock trading problems. Following the synthetic portfolio optimization setup, we study the six considered methods for 21-day (one month) stock trading in four different stock sectors using the daily stock price data from Yahoo finance between January 4, 2006 and April 1, 2022. For factors, we follow existing works (Aboussalah, 2020; De Prado, 2018; Dixon et al., 2020) and consider economic indexes, technical analysis indexes, and sector-specific features such as oil prices, gold prices, and related ETF prices, leading to around 30 factors for each sector. In each sector we select 10 stocks according to the availability and trading volume in the considered time range (Appendix L.1). The training, validation, and testing data are constructed using rolling windows (Appendix L.3). Table 3 reports the achieved average utility of each method under the selected hyperparamters. FaLPO achieves the highest average utility in all four sectors.

Next, we conduct the training-tuning-testing procedure above with $\gamma \in \{5, 7, 9\}$, and report the returns of FaLPO in each quarter in Figure 4. Recall that a smaller $\gamma$ corresponds to taking more risk. This is consistent with the observation in Figure 4 that the smaller the $\gamma$ the bigger the return but the larger the fluctuations. Also, the return fluctuates and drops around late 2018 and early 2020. The former corresponds to the abrupt bear market at the end of 2018, and the latter is consistent with the time period that COVID-19 bursts. Under these two time periods, the financial market was especially noisy and unpredictable. We also implement sensitivity analysis on $\lambda$ in Appendix L.4 and observe that a non-zero small $\lambda$ works well in practice.

## 6 EPILOGUE

**Conclusion** This work proposes FaLPO, a new decision-making framework for portfolio optimization with stochastic factors. By using continuous-time finance models to regularize policy learning, FaLPO is able to handle high noise and complex effects in financial data. We demonstrate FaLPO's benefits both theoretically and empirically. We focus on policy learning and defer more advanced feature engineering methods to future work.

**Limitations** FaLPO has two potential limitations. First, while we show the extension of FaLPO to problems beyond portfolio optimization, FaLPO is not applicable when there is no suitable parametric model to derive the optimal policy functional form. In such cases, the model-regularized policy learning of FaLPO cannot be implemented. Second, the performance of FaLPO still relies on good features ($Y_t$) to generate factors ($X_t$). In the presence of unpredictable market events (like COVID-19), or when the features do not contain any useful signals (like the Merton case in Appendix K.4.4), additional caution needs to be taken when applying FaLPO.

**Reproducibility** The assumptions and proof details are provided in Appendices I and J. The experiment implementation details are reported in Appendices K and L.

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

APPENDIX

# A  RELATED LITERATURE

In this section, we discuss related literature.

## A.1  CONTINUOUS-TIME FINANCE MODELS

Existing strategies for solving portfolio optimization using continuous-time finance models

can be loosely summarized as performing three steps:

1. Choosing the model for the dynamics, i.e. the type of stochastic differential equation (SDE).
2. Estimating the parameter of the selected model (which is also referred to as model fitting, model identification, or calibration).
3. Solving for the optimal policy under the estimated model.

The third step leverages stochastic optimal control tools (Fleming & Rishel, 1975; Fleming & Mitter, 1982; Fleming & Soner, 2006; Yong & Zhou, 1999).

Finding and estimating an appropriate model for stochastic optimal control requires significant domain knowledge. For instance, in finance, the modeler must specify both which features are relevant and how they affect stock prices (Fama & French, 1992). If not every relevant factor is correctly specified, optimal control can hardly lead to good performances. As a result, in stock trading, control methods would hand pick three to five economic indices as the factors and assume they follow a simple (often linear) SDE. But indeed trading can benefit from much richer datasets including related option prices, technical indicators, and interest rates (Aboussalah, 2020; De Prado, 2018; Dixon et al., 2020; Mehtab & Sen, 2019).

Further, even with a correctly specified model and factors, likelihood-based estimation for SDE control models can be very challenging (Phillips & Yu, 2009). As a result, methods like Aït-Sahalia (2008); Ait-Sahalia & Kimmel (2010) seek to replace the exact likelihood with other likelihood-based objective functions, while maintaining theoretical guarantees. However, the proposed objective function needs to be derived for each specific problem, and the derivation can be challenging. Other methods like Fasen (2013); Holỳ & Tomanová (2018) rely on more specific parametric or low-dimensional setups. To alleviate these issues, our framework extends the existing continuous-time finance models by allowing for a flexible and generalized definition of stochastic factor dynamics. Further, we simultaneously conduct policy learning and model calibration in an RL manner, with a square-loss objective that avoids the calculation of an exact likelihood.

## A.2  REINFORCEMENT LEARNING

RL aims to conduct the aforementioned three steps by (i) relying more on data (ii) in an end-to-end fashion. Methods like model-free RL assume no parametric forms on the dynamics, and directly learn the optimal policy while explicitly learning the model (step 1) and estimating the parameters (step 2).

**Discrete-Time Model-Free RL**   There exist many discrete-time model-free RL methods (Sutton & Barto, 2018). In this category, deep deterministic policy gradient (DDPG) is the most relevant one, and is empirically most widely used for portfolio optimization (Hambly et al., 2021). The reason is twofold. First of all, DDPG is a policy-gradient based method, and thus can naturally handle continuous states and actions in portfolio optimization with simple procedures. Second, DDPG learns a deterministic policy instead of a stochastic one like Haarnoja et al. (2018). This characteristic is especially important in portfolio optimization where the policy learning goal is a deterministic policy since the cost of a stochastic policy is extremely expensive.

**Continuous-Time Model-Free RL**   Continuous-time model-free RL (Wang et al., 2018; Doya, 2000; Munos, 2006) aims to solve for a continuous-time policy. However, such methods do not use or assume any SDE structure, and thus struggle with the common open questions in model-free RL like poor stability and sample complexity. As one example, path integral methods stem from the

theoretical result that the value function of a type of continuous-time decision-making problems can be expressed in closed form as a Feyman-Kac path integral (Fleming & Rishel, 1975; Kappen, 2005). A series of control/RL methods follow the rationale of optimizing the policy to maximize such an integral. Specifically, Theodorou et al. (2010) propose an open-loop control strategy; Kappen & Ruiz (2016) builds RL with importance sampling; Chebotar et al. (2017a;b); Stulp & Sigaud (2012) combine path integral with other model-based or model-free RL methods. However, the core derivation only holds for decision-making satisfying Kappen & Ruiz (2016, Equation (1)), which is equivalent to assuming that the action does not affect the randomness in decision-making. Such an assumption is limiting, and does not hold for portfolio optimization, where how to allocate the wealth in order to minimize the risk is key to a successful policy.

**Model-based RL and RL with Representation Learning**   Model-based RL and RL with representation learning are two active research areas but without a clear general state-of-the-art (Bharadhwaj et al., 2022; Eysenbach et al., 2021; Trabucco et al., 2022; Janner et al., 2019; Deisenroth & Rasmussen, 2011; Nagabandi et al., 2018; Laskin et al., 2021; Lee et al., 2020; Watter et al., 2015; Chebotar et al., 2017a; Hafner et al., 2019; Kim et al., 2019). The closest to FaLPO are those that learn an explicit representation of a latent variable like Lee et al. (2020); Mhammedi et al. (2020). But such methods are unable to leverage continuous-time finance models for portfolio optimization.

**Bayesian RL**   Our proposed framework is also related to Bayesian models (Ghavamzadeh et al., 2015; Rawlik et al., 2012), if we treat the learned representation of factors as the hidden variable. Strictly formulating an NSFM as a Bayesian model requires assumptions specifying the conditional distributions, and thus requires more domain knowledge. The optimization of Bayesian methods is also more challenging.

**RL for Stock Trading**   Various efforts have been made on applying RL to stock trading (Corazza & Bertoluzzo, 2014; Hambly et al., 2021; Nan et al., 2022; Xiong et al., 2018; Guan & Liu, 2021; Liu et al., 2021; Hu et al., 2018; Yu et al., 2019). However, these methods focus more on feature selection or empirical performance-improving techniques. Methodologically, they do not take advantage of continuous-time finance models.

### A.3   EMPIRICAL RISK MINIMIZATION

Another related area is Empirical Risk Minimization (ERM) (Vapnik, 1992). ERM studies the minimization of an objective function using the averages over training data to construct an empirical loss function. Recent work connected ERM with simulation-based and data-based offline decision-making methods (Reppen & Soner, 2020). More specifically, when the random input is observable and unaffected by actions, and a training set is available, the decision-making problem can be formulated as an ERM problem. As a result, the portfolio optimization may be reformulated as an ERM extension.

## B   OTHER OBJECTIVE FUNCTIONS

Note that the goal of portfolio optimization is to maximize the return while minimize or constrain the risk. In practice, one can use different objective functions for such a goal, like mean-variance objective (Hambly et al., 2021), Sharpe ratio, and so on. In this work, we consider utility maximization with power utility and exponential utility. The proposed method also works with other objective functions, as long as we can derive (part of) the optimal policy structure. Note that the selection among these objective functions is more a user-preference question.

## C    GRADIENT ESTIMATES

In this section, we discuss the gradient estimation for both $V$ and $L$. Assume that we collect $B$ independent trajectories for $S_t$ and $Y_t$, denoted as

$$\mathbb{D} := \{(s_{0,k}, y_{0,k}), (s_{\Delta t,k}, y_{\Delta t,k}), (s_{2\Delta t,k}, y_{2\Delta t,k}), \cdots (s_{M\Delta t,k}, y_{M\Delta t,k})\}_{k=1}^{B}.$$

Then, the gradient estimate for $V(\theta_\phi, \theta_\pi)$ is defined as

$$\bar{\nabla}V(\theta_\phi, \theta_\pi) := \frac{1}{B}\sum_{k=1}^{B} \tilde{\nabla}V_k(\theta_\phi, \theta_\pi) \text{ with } \tilde{\nabla}V_k(\theta_\phi, \theta_\pi) := \nabla_{\theta_\phi, \theta_\pi} U(z_{T,k}^{\pi(\cdot; \theta_\phi, \theta_\pi), \Delta t}).$$

The terminal wealth in the trajectory $k$ under the policy where $\pi(\cdot; \theta_\pi)$ is denoted as $z_{T,k}^{\pi(\cdot; \theta_\pi), \Delta t}$ with

$$z_{T,k}^{\pi(\cdot; \theta_\pi), \Delta t} := z_0 + \sum_{m=1}^{M} z_{m-1}\left[\sum_{i=1}^{d_S} \pi^i(m\Delta t, y_{m\Delta t}; \theta_\phi, \theta_\pi)\frac{s_{(m+1)\Delta t}^i - s_{m\Delta t}^i}{s_{m\Delta t}^i}\right].$$

Next, we consider the gradient of $L$:

$$\bar{\nabla}L(\theta_\phi, \theta_S) := \frac{1}{B}\sum_{k=1}^{B} \tilde{\nabla}L_k(\theta_\phi, \theta_S).$$

Specifically for likelihood and negative mean square loss, we have

$$\tilde{\nabla}L_{Likelihood,k}(\theta_\phi, \theta_S) := \frac{1}{M}\sum_{m=0}^{M-1} \nabla_{\theta_\phi, \theta_S} \log(\mathbb{P}(s_{(m+1)\Delta t,k}, \phi(y_{(m+1)\Delta t,k}; \theta_\phi)$$
$$\mid s_{m\Delta t,k}, \phi(y_{m\Delta t,k}; \theta_\phi); \theta_S)),$$

$$\tilde{\nabla}L_{NMSL,k}(\theta_\phi, \theta_S) := -\frac{1}{M}\sum_{m=0}^{M-1}\sum_{i=1}^{d_S} \nabla_{\theta_\phi, \theta_S}\bigg(\log(s_{(m+1)\Delta t,k}^i) - \log(s_{m\Delta t,k}^i)$$
$$- \mathbb{E}\bigg[\int_{m\Delta t}^{(m+1)\Delta t} f_S^j(X_s; \theta_S)$$
$$- \frac{1}{2}\sum_{i=1}^{d_S}(g_S^{ij}(X_s; \theta_S))^2 ds \bigg| s_{m\Delta t,k}, \phi(y_{m\Delta t,k}; \theta_\phi)\bigg]\bigg)^2.$$

As a result, in each iteration, we collect $B$ trajectories to estimate the gradient of $H(\theta_\phi, \theta_\pi)$.

## D    A PRIMER ON STOCHASTIC DIFFERENTIAL EQUATIONS (SDEs)

We provide a general formulation of SDEs with two examples.

### D.1    FORMULATION OF SDEs

SDEs are a generalization of ordinary differential equations to dynamic systems influenced by random fluctuations. The structure of the randomness can in principle be quite general, such as with jump processes where the state evolution is no longer continuous (Tankov, 2003). Although our method can be generalized to all SDEs, we restrict ourselves to practical settings where the source of randomness is a Brownian motion.

Let $W_t$ be a multi-dimensional independent standard Brownian motion. For a random process $S_t$, an SDE is typically expressed using a differential form as

$$dS_t = f(S_t)\,dt + g(S_t)\,dW_t, \text{ or } S_t = S_0 + \int_0^t f(S_t)\,dt + \int_0^t g(S_t)\,dW_t, \tag{8}$$

where $f(\cdot)$ and $g(\cdot)$ are functions of $S_t$. The stochastic integral $\int_0^t g(S_t)dW_t$ is the accumulation of influence to the state due to the noise. We refer the reader to Karatzas & Shreve (1987) for details on the construction of stochastic integrals and SDE theory. Important here is that Equation (8) defines the transition of $S_t$ in an infinitesimal time step. The drift coefficient $f(S_t)$ characterizes the deterministic part of the change of $S_t$, and the diffusion coefficient $g(S_t)$ models the randomness in the transition of $S_t$.

## D.2 EXAMPLES

As concrete examples, we discuss two families of SDEs widely used in finance, economics, and biology: Geometric Brownian motion (GBM) and Ornstein–Uhlenbeck (OU) processes (Merton et al., 1973; Vasicek, 1977; Bartoszek et al., 2017; Blomberg et al., 2020; Rohlfs et al., 2014). The OU structure appears in both applications below, and the financial application uses GBM as a base, but extends it with OU drift coefficients. The two types of SDEs are given by

$$\text{GBM: } \frac{dS_t}{S_t} = \mu\, dt + \sigma\, dW_t, \qquad \text{OU: } dS_t = \mu S_t\, dt + \sigma\, dW_t,$$

where $\frac{d\mathbf{S}_t}{\mathbf{S}_t} := \left\{ \frac{dS_{it}}{S_{it}} \right\}$ denotes the component-wise division of $S_t$, and the matrices $\mu$ and $\sigma$ define the drift and diffusion coefficients.

We refer the interested reader to Fleming & Soner (2006) for more information on these topics. We now briefly formulate two classic stochastic optimal control models for decision-making with stochastic factors. The stochastic factors appear as the drift coefficients of other state variables and are themselves modeled as SDEs.

## D.3 ITÔ'S FORMULA

Itô's Formula is a fundamental analytical tool for SDEs, and crucial for their analysis. We only provide a simple version here, which is sufficient for our analysis. A more general and rigorous statement with assumptions and proof of Itô's formula and integral can be found in Karatzas & Shreve (1987, Theorem 3.3).

LEMMA D.1 (Itô's Formula)  Consider a twice differentiable function $G$, and $S_t$ following

$$dS_t = f(S_t)\, dt + g(S_t)\, dW_t.$$

Then, we have

$$dG(t, S_t) = \left\{ \frac{\partial G}{\partial t} + \left( \frac{\partial G}{\partial S_t} \right)^\top f(S_t) + \frac{1}{2} \operatorname{Tr} \left[ g(S_t)^\top \frac{\partial^2 G}{\partial S_t^2} g(S_t) \right] \right\} dt + \left( \frac{\partial G}{\partial S_t} \right)^\top g(S_t)dW_t.$$

LEMMA D.2  For a suitable bounded process $S_t$, the Itô integral $\int_0^t S_t dW_t$ satisfies:

$$\mathbb{E}\left[ \int_0^t S_t dW_t \right] = 0, \qquad \mathbb{E}\left[ \left( \int_0^t S_t dW_t \right)^2 \right] = \mathbb{E}\left[ \int_0^t S_t^2 dt \right].$$

The latter is also referred to as Itô's isometry

## E  MODEL CALIBRATION

We discuss two model calibration loss functions, log-likelihood and negative mean square loss.

## E.1  LOG-LIKELIHOOD

We can use log-likelihood as $L$ for model calibration. The log-likelihood of SDEs is derived in a sequential manner. Specifically, for (2) the log-likelihood is derived as

$$L_{Log-Likelihood}(\theta_\phi, \theta_S) := \mathbb{E}[\log(\mathrm{P}_{\theta_S}(S_{t+\Delta t}, \phi(Y_{t+\Delta t}; \theta_\phi) \mid S_t, \phi(Y_t; \theta_\phi)))], \qquad (9)$$

where $\mathrm{P}_{\theta_S}$ denotes the conditional likelihood according to (2) but with parameter $\theta_S$ instead of $\theta_S^*$. Then, in specific models, one can derive $\mathrm{P}_{\theta_S}(S_{t+\Delta t}, \phi(Y_{t+\Delta t}) \mid S_t, \phi(Y_t); \theta)$ (Phillips, 1972; Holỳ & Tomanová, 2018; Beskos & Roberts, 2005) or the approximation of it (Ait-Sahalia & Kimmel, 2010).

## E.2 NEGATIVE MEAN SQUARE LOSS

For the SDE system (2), one can also use a negative mean square loss (NMSL) as the calibration objective. To derive this loss function, we first drive the dynamics of log price by applying Itô's Formula Lemma D.1 to (2):

$$d\log(S_t^i) = f_S^i(X_t;\theta_S)dt - \frac{1}{2}\sum_{j=1}^{d_W}(g_S^{ij}(X_s;\theta_S)])^2 dt + \sum_{j=1}^{d_W} g_S^{ij}(X_s;\theta_S)]dW_t^j. \tag{10}$$

Then, combined with Lemma D.2, under proper assumptions of Lemma D.2, we pose expectation over both sides of the above equation, and derive

$$\mathbb{E}[d\log(S_t^i)] = f_S^i(X_t;\theta_S)dt - \frac{1}{2}\sum_{j=1}^{d_W}(g_S^{ij}(X_s;\theta_S)])^2 dt.$$

In words, the expectation of the log price change in an infinitesimal time is $f_S^i(X_t;\theta_S)dt - \frac{1}{2}\sum_{j=1}^{d_W}(g_S^{ij}(X_s;\theta_S)])^2 dt$. Therefore, one can estimate the parameter $\theta_S$ by minimizing the mean square loss between the log price change and the expected log price change:

$$L_{NMSL}(\theta_S) := -\mathbb{E}\bigg[\sum_{i=1}^{d_S}\bigg(\log(S_{t+\Delta t}^i) - \log(S_t^i) - \mathbb{E}\bigg[\int_t^{t+\Delta t} f_S^i(X_s;\theta_S)$$
$$- \frac{1}{2}\sum_{j=1}^{d_W}(g_S^{ij}(X_s;\theta_S))^2 ds\bigg|S_t, X_t\bigg]\bigg)^2\bigg]. \tag{11}$$

It can be easily proved that the true data generating SDE parameter satisfies

$$\theta_S^* \in \arg\max L_{NMSL}(\theta_S).$$

Further, if we take $X_t = \phi(Y_t;\theta_\phi)$ and parameterize the objective as $L_{NMSL}(\theta_\phi,\theta_S) := -\mathbb{E}\bigg[\sum_{i=1}^{d_S}\big[\log(S_{t+\Delta t}^i) - \log(S_t^i) - \phi^i(Y_t;\theta_\phi)\Delta t - \theta_S^i\big]^2\bigg]$, we can prove

$$\theta_\phi^*, \theta_S^* \in \arg\max L_{NMSL}(\theta_\phi,\theta_S).$$

Note that in practice it can be very hard to calculate the expectation $\mathbb{E}\big[\int_t^{t+\Delta t} f_S^i(X_s;\theta_S) - \frac{1}{2}\sum_{j=1}^{d_W}(g_S^{ij}(X_s;\theta_S))^2 ds\big|S_t, X_t\big]$. Therefore, when $\Delta t$ is small, we replace the conditional expectation via $\mathbb{E}\big[f_S^i(X_S;\theta_S)\Delta t - \frac{1}{2}\sum_{j=1}^{d_W}(g_S^{ij}(X_S;\theta_S))^2\Delta t\big|S_t, X_t\big]$. Accordingly the calibration objective is defined as

$$L_{NMSL}(\theta_S) \approx -\mathbb{E}\bigg[\sum_{i=1}^{d_S}\bigg(\log(S_{t+\Delta t}^i) - \log(S_t^i) - f_S^i(X_s;\theta_S)\Delta t + \frac{1}{2}\sum_{j=1}^{d_W}(g_S^{ij}(X_s;\theta_S))^2\Delta t\bigg)^2\bigg]. \tag{12}$$

## E.3 OTHER MODEL CALIBRATION OBJECTIVE

Another potential model calibration objective following the same rationale as (11) is

$$L_{NMSL-X}(\theta_\phi,\theta_S) := -\mathbb{E}\bigg[\sum_{i=1}^{d_X}\bigg(\phi(Y_{t+\Delta t};\theta_\phi)^i - \phi(Y_t;\theta_\phi)^i$$
$$- \mathbb{E}\bigg[\int_t^{t+\Delta t} f_X^i(\phi(Y_s;\theta_\phi)^i;\theta_S)ds\bigg|\phi(Y_t;\theta_\phi)^i\bigg]\bigg)^2\bigg],$$

which is derived using the conditional expectation of $X_{t+\Delta}$ given $X_t$. However, the true data-generating parameter $(\theta_\phi^*, \theta_S^*)$ is not a maximal point of $L_{NMSL-X}$. To more clearly see this, this loss function encourages the representation function $\phi$ to take a constant output so that $X_t$ is constant over time with $f_X^i(X_s;\theta_S) = 0$, and $L_{NMSL-X}(\theta_\phi) = 0$. We also try this loss in experiments, and it leads to poor validation and test performances.

# F APPLICATIONS OF FALPO TO DIFFERENT STOCHASTIC FACTOR MODELS IN CONTINUOUS-TIME FINANCE

FaLPO can be used with many stochastic factor models in continuous-time finance models. In this section, we discuss the Merton model (Appendix F.1), Kim–Omberg (Appendix F.2) and EVE (Appendix F.3).

## F.1 MERTON MODEL

Merton model (Merton, 1969) is a classic setup for portfolio optimization. It studies the allocation of capital across a set of financial assets in order to maximize profits and minimize risks.

### F.1.1 MODELING

Consider $p$ risky assets with prices $S_t = \{S_t^i\}_{i=1}^p$ and an additional risk-free money market account with, for simplicity, zero interest rate of return (like cash). The Merton model does not include factors. The dynamics for asset prices is formulated as

$$\frac{dS_t^i}{S_t^i} = \mu \, dt + \sigma dW_t. \tag{13}$$

The parameters $\mu$, $\sigma$ are $p \times p$ matrices, with $\sigma$ denoting the volatility of assets. Further, we use $\tilde{Z}_t^{\tilde{\pi}}$ to denote the wealth at time point $t$ under the continuous-time policy $\tilde{\pi}$. Under the famous and widely used self-financing assumption (Björk, 2009), we have

$$\frac{d\tilde{Z}_t^{\tilde{\pi}}}{\tilde{Z}_t^{\tilde{\pi}}} = \tilde{\pi}_t \mu dt + \tilde{\pi}_t \sigma dW_t.$$

An investor's goal is to maximize the expected utility of capital $U(Z_T)$ at some future time point $T$:

$$\max_{\tilde{\pi}_t} \mathbb{E}_{\tilde{\pi}_t} \left[ U(\tilde{Z}_T^{\tilde{\pi}}) | Z_0 = z, S_0 = s \right]. \tag{14}$$

Negative values in the policy output are allowed, meaning the agent can short any asset.

### F.1.2 POLICY FUNCTIONAL FORM

For the Merton model, $\Pi$ in (3) can be explicitly derived.

LEMMA F.1 (Policy Functional Form for Merton Model) For a Merton model defined in (13), under common assumptions, the optimal policy for portfolio optimization (14), with power utility follows

$$\tilde{\pi}^* = \mu(\sigma\sigma^\top)^{-1}.$$

The optimal policy for portfolio optimization (14) with exponential utility follows

$$\tilde{\pi}^* = \frac{\mu(\sigma\sigma^\top)^{-1}}{\tilde{Z}_t^{\tilde{\pi}^*}}.$$

*Proof.* Lemma F.1 is a classic result in continuous-time finance, proposed in Merton (1969). □

In words, in a Merton model, the optimal policy is independent of time, stock prices, features, and factors. The optimal investment strategy is to keep a constant fraction or amount of wealth in each asset all along, depending on the choice of utility function. Let $\theta_\pi$ be a $d_S \times 1$ parameter vector. According to Lemma F.1, in FaLPO, we parameterize the candidate policy function as

$$\pi(t, S_t, Z_t, Y_t; \theta_\phi, \theta_\pi) = \Pi(t, S_t, Z_t, \phi(Y_t; \theta_\phi); \theta_\pi) = \theta_\pi$$

for power utility, and

$$\pi(t, S_t, Z_t, Y_t; \theta_\phi, \theta_\pi) = \Pi(t, S_t, Z_t, \phi(Y_t; \theta_\phi); \theta_\pi) = \frac{\theta_\pi}{Z_t}$$

for exponential utility.

### F.1.3 MODEL CALIBRATION

According to the Merton model formulation, there exist no factors affecting the evolution of stock prices. Therefore, we do not add the model calibration objective in FaLPO for Merton problem.

## F.2 KIM–OMBERG

Kim–Omberg model (Kim & Omberg, 1996) is a standard model for portfolio optimization with predictable asset returns, which has been discussed extensively in the empirical literature (Welch & Goyal, 2008; Muhle-Karbe et al., 2017).

### F.2.1 MODELING

In Kim–Omberg model, the stock dynamics are formulated as

$$\frac{dS_t^i}{S_t^i} = X_t^i \, dt + \sum_{j=1}^{d_W} \sigma^{ij} \, dW_t^j$$

$$dX_t = \mu(\omega - X_t) \, dt + v \, dW_t.$$

The portfolio optimization goal is formulated as

$$\max_{\tilde{\pi}_t} \tilde{V}(\tilde{\pi}_t) \text{ with } \tilde{V}(\tilde{\pi}_t) := \mathbb{E}_{\tilde{\pi}_t}[U(\tilde{Z}_t^{\tilde{\pi}})|X_0 = x, Z_0 = z, S_0 = s]. \tag{15}$$

### F.2.2 POLICY FUNCTIONAL FORM

An optimal policy function is derived in Lemma F.2

LEMMA F.2 Under common assumptions in Kim & Omberg (1996); Herzog et al. (2004), an optimal policy functional form for (15) with power utility is derived as

$$\tilde{\pi}^* = \frac{1}{1-\gamma}(\sigma\sigma^\top)^{-1}\big[X_t + \sigma v^\top(k_3(t)X_t + k_2(t))\big],$$

where $k_2(t)$ and $k_3(t)$ satisfy

$$\frac{dk_1(t)}{dt} + \frac{1}{2}\operatorname{Tr}\left\{v^\top(k_2(t)k_2(t)^\top + k_3(t))v\right\} + (\mu\omega)^\top k_2(t) - \frac{\gamma}{2(\gamma-1)}(k_2(t)^\top vv^\top k_2(t)) = 0,$$

$$\frac{dk_2(t)}{dt} + k_3(t)vv^\top k_2(t) - \mu^\top k_2(t) + k_3(t)\mu\omega - \frac{\gamma}{\gamma-1}((\sigma\sigma^\top)^{-1}\sigma v k_2(t) + k_3(t)vv^\top k_2(t)) = 0,$$

$$\frac{dk_3(t)}{dt} + k_3(t)vv^\top k_3(t) - k_3(t)\mu - \mu^\top k_3(t)$$
$$- \frac{\gamma}{\gamma-1}((\sigma\sigma^\top)^{-1} + (\sigma\sigma^\top)^{-1}\sigma v^\top k_3(t) + k_3(t)v\sigma^\top(\sigma\sigma^\top)^{-1} + k_3(t)vv^\top k_3(t)) = 0,$$

with $k_1(T) = 0$, $k_2(T) = 0$, and $k_3(T) = 0$. Note that $k_1(t)$ is a scalar, $k_2(t)$ is a $d_S \times 1$ vector, and $k_3(t)$ is a $d_X \times d_X$. And the ODE of $k_3(t)$ is the famous matrix Ricatti equation.

Similarly, an optimal policy functional form for (15) with exponential utility is derived as

$$\tilde{\pi}_t^* = (\sigma\sigma^\top)^{-1}\frac{1}{-\gamma^2 Z_t}\left[X(t) + \sigma v^\top(k_3(t)X_t + k_2(t))\right],$$

where $k_2(t)$ and $k_3(t)$ satisfy

$$\frac{dk_1(t)}{dt} + \frac{1}{2}\operatorname{Tr}\left\{v^\top(k_2(t)k_2(t)^\top + k_3(t))v\right\} + (\mu\omega)^\top k_2(t) - \frac{1}{2}(k_2(t)^\top vv^\top k_2(t)) = 0,$$

$$\frac{dk_2(t)}{dt} + k_3(t)vv^\top k_2(t) - \mu k_2(t) + k_3(t)\mu\omega - ((\sigma\sigma^\top)^{-1}\sigma v k_2(t) + k_3(t)vv^\top k_2(t)) = 0,$$

$$\frac{dk_3(t)}{dt} + k_3(t)vv^\top k_3(t) - k_3(t)\mu - \mu^\top k_3(t)$$
$$- ((\sigma\sigma^\top)^{-1} + (\sigma\sigma^\top)^{-1}\sigma v^\top k_3(t) + k_3(t)v\sigma^\top(\sigma\sigma^\top)^{-1} + k_3(t)vv^\top k_3(t)),$$

with $k_1(T) = 0$, $k_2(T) = 0$, and $k_3(T) = 0$.

### F.2.3 MODEL CALIBRATION

Following the derivation in Appendix E.2, the negative mean square loss for Kim–Omberg model can be derived as $L(\theta_\phi, \theta_S) := -\mathbb{E}\left[\sum_{i=1}^{d_S} \left[\log(S_{t+\Delta t}^i) - \log(S_t^i) - \phi^i(Y_t; \theta_\phi)\Delta t - \theta_S^i\right]^2\right]$, where in this case $\theta_S$ is a $d_S \times 1$ vector.

### F.3 EVE MODEL WITH STOCHASTIC MARKOVIAN FACTORS

We take EVE model (Avanesyan et al., 2018) with stochastic Markovian factors as another example.

#### F.3.1 MODELING

We first detail the modeling of EVE, which formulates the dynamics of asset prices by

$$
\begin{aligned}
\frac{dS_t^i}{S_t^i} &= \mu_i(X_t; \theta_S)dt + \sum_{j=1}^{d_W} \sigma_{ji}(X_t; \theta_S)dW_t^j, \quad i = 1, 2, \cdots, d_S, \\
dX_t &= \left(M^\top X_t + \omega\right)dt + \kappa(X_t; \theta_S)^\top dB_t, \\
B_t &= \rho^\top W_t + A^\top W_t^\perp.
\end{aligned}
\tag{16}
$$

We use $W_t$ and $W_t^\perp$ to denote two sets of independent Brownian motions. $\rho$ denotes a correlation matrix with components $\rho^{ij} \in [-1, 1]$. Therefore, $B$ is indeed another Brownian motion. We let $\mu$, $\sigma$, and $\kappa$ be parametric functions, with $\theta_S$ denoting all the parameters. The SDE parameters include all the parameter matrices and $\boldsymbol{\beta}$'s. Again, with $Z_t$ as the wealth, we aim to maximize the power utility at the terminal time $T > 0$ as the performance objective:

$$
\tilde{V}(\tilde{\pi}_t) = \mathbb{E}_{\tilde{\pi}}\left[\frac{(\tilde{Z}_T^{\tilde{\pi}})^{1-\gamma}}{1-\gamma}\right].
$$

EVE model poses further assumptions on (16).

ASSUMPTION F.3  $M$ has non-negative off-diagonal entries and $\omega \in [0, \infty)^k$. Further, we assume that there exist $\lambda(x)$, $\Lambda$ and $L$, and $N$ such that $\mu(\cdot)$, $\sigma(\cdot)$, $\kappa(\cdot)$ and $\rho$ satisfy

$$
\begin{aligned}
\lambda(x)^\top \lambda(x) &= \mu(x)^\top \sigma(x)^{-1}\left(\sigma(x)^{-1}\right)^\top \mu(x) = \Lambda^\top x \\
\kappa(x)^\top \kappa(x) &= diag(L_1 x_1, L_2 x_2, \cdots, L_k x_k), \text{ with } L_1, L_2, \cdots, L_k \geq 0 \\
\Gamma \kappa(x)^\top \rho^\top \lambda(x) &= N^\top x.
\end{aligned}
\tag{17}
$$

The conditions in Assumption F.3 are necessary for the process of $X_t$ to be $[0, \infty)^k$-valued and affine. Under these conditions, the SDE in (16) has a unique weak solution which is affine and takes values in $[0, \infty)^k$ (Filipović & Mayerhofer, 2009). Further, the EVE model requires the following two assumptions:

ASSUMPTION F.4 (Assumption 2.2 in Avanesyan et al. (2018))  The functions $\mu : \mathbb{R}^{d_x} \to \mathbb{R}^{d_S}$, $\sigma : \mathbb{R}^{d_x} \to \mathbb{R}^{d_W \times d_S}$ are continuous. More over, the columns of $\rho$ belong to the range of left-multiplication by $\sigma(x)$ for all $x \in \mathbb{R}^{d_x}$.

ASSUMPTION F.5 (EVE Condition in Avanesyan et al. (2018))  For some $p \in [0, 1]$,

$$
\rho^\top \rho = pI,
$$

where $I$ is the identity matrix. Note that when $p = 1$, $\rho$ is a vector and thus we define $p := \rho^\top \rho$.

#### F.3.2 CONCRETE EXAMPLE

We consider a more concrete example of EVE satisfying the formulation and assumptions in Appendix F.3.1. Specifically, we use $D_\mu$, $D_\sigma$, and $D_\lambda$ to denote diagonal matrices. Further, let $D(x)$ denote the diagonal matrix whose diagonal is $x$. Also, we use $x^{\circ k}$ for any $k \in \mathbb{R}$ to denote the component-wise power operation (Hadamard power).

Then, we define

$$\mu(x) := D_\mu x^{\circ\frac{3}{2}}$$
$$\sigma(x) := D_\sigma D(x)$$
$$\kappa(x) := \rho^{-1} D_\kappa D(x^{\circ\frac{1}{2}}).$$

Then, we have

$$\lambda(x) = D_\sigma^{-1} D_\mu x^{\circ\frac{1}{2}}.$$

Further, we pose

$$\rho^\top \rho = \rho\rho^\top = I.$$

Then,

$$\lambda(x) = \left(\sigma(x)^{-1}\right)^\top \mu(x) = D_\sigma^{-1} x^{\circ\frac{1}{2}} \text{ and } \lambda(x)^\top \lambda(x) = D_\sigma^{-2} x,$$
$$\kappa(x)^\top \kappa(x) = D_\kappa D(x^{\circ\frac{1}{2}})(\rho^{-1})^\top \rho^{-1} D_\kappa D(x^{\circ\frac{1}{2}}) = D_\kappa^2 D(x),$$
$$\Gamma\kappa(x)^\top \rho^\top \lambda(x) = \Gamma D(x^{\circ\frac{1}{2}}) D_\kappa (\rho^{-1})^\top \rho^\top D_\sigma^{-1} x^{\circ\frac{1}{2}} = \Gamma D(x^{\circ\frac{1}{2}}) D_\kappa D_\sigma^{-1} x^{\circ\frac{1}{2}} = \Gamma D_\kappa D_\sigma^{-1} x.$$

Such a setup is shown satisfies (16).

### F.3.3    Policy Functional Form

The policy functional form of the EVE model with Markovian stochastic factors ca be derived as:

LEMMA F.6   Under the assumptions in Appendix F.3.1 and Appendix F.3.2, the optimal policy function follows:

$$\pi_t^* = \frac{1}{\gamma}\sigma(X_t)^{-1}\left(\lambda(X_t) + q\rho\kappa(X_t)k_2(t)^\top\right),$$

with

$$\frac{dk_2^i(t)}{dt} + \frac{1}{2}L_i(k_2(t)^i)^2 + \sum_{j=1}^{d_X}(M+N)^{ij}k_2(t)^j + \frac{\Gamma}{2q}\Lambda^i = 0, \quad i = 1, 2, \cdots, d_X$$

$$\frac{dk_1(t)}{dt} + \omega^\top k_2(t) = 0.$$

For the specific example in F.3.2, in FaLPO, we parameterize the candidate policy as

$$\pi(t, Y_t, Z_t; \theta_\phi, \theta_\pi) = \Pi(t, Y_t, Z_t; \theta_\phi, \theta_\pi) = \phi(Y_t)^{\frac{1}{2}} k(t; \theta_\pi).$$

### F.4    Model Calibration

Following the derivations in Section E.2, we can derive the calibration objective as:

$$L(\theta_S) := -\min_{\theta_S=(C_1,C_2)}\mathbb{E}\left\|\Delta\log(S_t) - C_1\phi(Y_t;\theta_\phi)^{\circ\frac{3}{2}}\Delta t - C_2\phi(Y_t;\theta_\phi)^{\circ 2}\Delta t\right\|_2^2.$$

## G    Solutions of Riccati Differential Equations

According to the analysis in Section F, the optimal policy function is closely related to the solutions of Riccati differential equations, which also have closed-form solutions.

Specifically, with abuse of notation, let $A$, $B$ and $D$ be $p \times p$ matrices, and $X(t) : [0, T] \to \mathbb{R}^{p\times p}$ as a function of $t$ solving the following Riccati differential equation:

$$\frac{\partial dX(t)}{dt} = A^\top X(t) + X(t)A - X(t)BB^\top X(t) + D^\top D,$$
$$X(0) = X_0. \tag{18}$$

Following the analysis and assumptions in (Behr et al., 2019), the unique symmetric positive stabilizing solution of $X(t)$ follows:

$$X(t) = X_\infty - e^{t\hat{A}^\top}\left(X_\infty - X_0\right)\left[I - (X_L - e^{t\hat{A}}X_L e^{t\hat{A}^\top})(X_\infty - X_0)\right]^{-1}e^{t\hat{A}},$$

where

$$\hat{A}X_L + X_L\hat{A}^\top + BB^\top = 0,$$

with $\hat{A} := A - BB^\top X_\infty$, and

$$0 = A^\top X_\infty + X_\infty A + D^\top D.$$

Note that with (18) we can further derive the policy functional forms without using neural networks to parameterize time-dependent fucntions.

## H    EXTENSION TO LINEAR QUADRATIC CONTROL (LQC)

The methodology of FaLPO can also be applied to decision-making problems other than portfolio optimization. To implement FaLPO, one needs to first construct a neural stochastic factor model combining factor representation with a continuous-time model. Then, the policy learning is conducted while leveraging policy functional form and model calibration. As an example, we implement FaLPO to linear quadratic control (LQC), and detail modeling (Section H.1), policy functional form (Section H.2) and model calibration (Section H.3).

### H.1    MODELING

We consider the problem of LQC (Sun & Yong, 2020) but with stochastic factor $X_t$ following an OU process. With slight abuse of notation, we use $S_t$ to denote the sate variable in this section:

$$dS_t = (BS_t + UA_t + X_t)\,dt + \sum_{j=1}^{d_W}D_j A_t\,dW_t^j,$$

$$X_t = \mu X_t\,dt + v\,dW_t,$$

(19)

where $B$, $U$, $\mu$, $v$, and $D_j$ are redefined as matrices with appropriate dimensions. With $A_t = \pi(\cdot)$ following the policy $\pi$, we aim to solve

$$\max_\pi V(\pi) \text{ with } V(\pi) := \mathbb{E}_\pi\left[\int_0^T \left[(QS_t)^\top S_t + (RA_t)^\top A_t\right]dt + (GS_T)^\top S_T\right],$$

(20)

with $Q$, $R$, and $G$ as known matrices with appropriate dimensions, and $T$ is terminal time. Further, we apply the modeling strategy in Section 3.1 and aim to learn the representation of stochastic factors from the available features $Y_t$:

$$X_t = \phi(Y_t; \theta_\phi^*).$$

### H.2    POLICY FUNCTIONAL FORM

By taking $\Xi_t$ (the combination of $S_t$ and $X_t$) as the state variables, we can reformulate the problem as a classic LQC problem:

$$d\Xi_t = B^\Xi\Xi_t + U^\Xi A_t + \sum_{j=1}^{d_W}(D_j^\Xi A_t + \beta_t^\Xi)dW_t^j,$$

with all the coefficients redefined. Then, under common assumptions in Sun & Yong (2020); Yong & Zhou (1999), it can be derived that the optimal policy satisfies:

$$\tilde{\pi}^*(t,\xi) = \Lambda^\Xi(K^\Xi(t))^{-1}(U^\Xi)^\top K^\Xi(t)\xi,$$

where

$$\Lambda^\Xi(K^\Xi(t)) = R + \sum_{j=1}^{d_W}(D_j^\top K^\Xi(t)D_j).$$

Also, $K^\Xi(t)$ solves the differential Riccati equation:

$$K^\Xi(t) = -e^{(B^\Xi)^\top(T-t)}G^\Xi e^{B^\Xi(T-t)}$$
$$- \int_t^T e^{(B^\Xi)^\top(T-\tau)}K^\Xi(\tau)^\top U^\Xi \big(\Lambda^\Xi(K^\Xi(\tau))^\top\big)^{-1}(U^\Xi)^\top K^\Xi(\tau)e^{(B^\Xi)^\top(T-\tau)}d\tau,$$

with

$$K^\Xi(T) = 0.$$

Therefore, we can formulate the candidate policy as

$$\pi(t, S_t, Y_t; \theta_\phi, \theta_\pi) = \Pi(t, S_t, Y_t; \theta_\phi, \theta_\pi) = k_1(t; \theta_\pi)S_t + k_2(t; \theta_\pi)\phi(Y_t; \theta_\phi). \tag{21}$$

### H.3 Model Calibration

According to (19) and following the derivation strategy in Appendix E.2, we can derive the negative mean square loss for LQC as

$$L(\theta_\phi, \theta_S) := \mathbb{E}\big[\|S_{t+\Delta t}S_t - \phi(Y_t; \theta_\phi)\Delta t - C_1 S_t - C_2 A_t\|_2^2\big], \tag{22}$$

with $\theta_S = \{C_1, C_2\}$.

As a summary, to apply FaLPO to LQC with stochastic factors, we parameterize candidate policies following (21) and maximize

$$(1 - \lambda V(\theta_\phi, \theta_\pi)) + \lambda L(\theta_\phi, \theta_S),$$

with V in (20) and L in (22).

## I Extended Results for Theorem 4.1

### I.1 Assumptions and Definitions

To start with, we consider stochastic factor models such that the optimal feedback admissible policy admits a functional form as

$$\tilde{\pi}_t^* = \Pi(t, X_t; \theta_{\tilde{\pi}}^*).$$

Also, we consider the $L$ function such that the true data generating parameters $\theta_S^*, \theta_\phi^*$ satisfy

$$\theta_S^*, \theta_\phi^* \in \underset{\theta_\phi, \theta_S}{\arg\max}\, L(\theta_\phi, \theta_S), \tag{23}$$

while other options for $L$ can also be empirically used in our method.

DEFINITION I.1 For a continuous-time policy $\tilde{\pi}_t := \Pi(t, S_t, Z_t, \phi(Y_t; \theta_\phi); \theta_{\tilde{\pi}})$, we define its value function as

$$\tilde{V}(\theta_\phi, \theta_\pi) := \mathbb{E}[U(Z_T^{\tilde{\pi}})].$$

Accordingly, we define the continuous-time version objective function as

$$\tilde{H}(\theta_\phi, \theta_{\tilde{\pi}}, \theta_S) := (1 - \lambda)\tilde{V}(\theta_\phi, \theta_{\tilde{\pi}}) + \lambda L(\theta_\phi, \theta_S).$$

DEFINITION I.2 For $t \in [m\Delta t, (m+1)\Delta t)$, define $\lfloor t \rfloor := m\Delta t$ and $\lfloor \tilde{\pi}_t^* \rfloor := \tilde{\pi}_{\lfloor t \rfloor}^*$.

DEFINITION I.3 For the continuous-time optimal policy $\tilde{\pi}^*$, we use $\lfloor \tilde{\pi}^* \rfloor$ to denote the piece-wise constant version $\tilde{\pi}^*$. We use $\tilde{Z}_t^{\lfloor \tilde{\pi}^* \rfloor}$ to denote the wealth process when implementing the optimal continuous-time policy $\tilde{\pi}(\cdot; \theta_\phi^*, \theta_\pi^*)$ in the piece-wise constant manner. Specifically,

$$d\tilde{Z}_t^{\lfloor \tilde{\pi}^* \rfloor} := \tilde{Z}_{\lfloor t \rfloor}^{\lfloor \tilde{\pi}^* \rfloor} \sum_{i=1}^{d_S} \frac{(\tilde{\pi}_{\lfloor t \rfloor}^*)^i dS_t^i}{S_{\lfloor t \rfloor}^i}$$

$$= \sum_{i=1}^{d_S} \left\{ \Pi^i(\lfloor t \rfloor, X_{\lfloor t \rfloor}; \theta_{\tilde{\pi}}^*) f_S^i(X_t; \theta_S^*) \frac{S_t^i}{S_{\lfloor t \rfloor}^i} \tilde{Z}_{\lfloor t \rfloor}^{\lfloor \tilde{\pi}^* \rfloor} dt \right.$$

$$\left. + \Pi^i(\lfloor t \rfloor, X_{\lfloor t \rfloor}; \theta_{\tilde{\pi}}^*) \sum_{j=1}^{d_W} g_S^{ij}(X_t; \theta_S^*) \frac{S_t^i}{S_{\lfloor t \rfloor}^i} \tilde{Z}_{\lfloor t \rfloor}^{\lfloor \tilde{\pi}^* \rfloor} dW_t^j \right\}.$$

Further, remember $\tilde{Z}_t^{\tilde{\pi}^*}$ is used to denote the continuous-time wealth process under the policy $\tilde{\pi}^*$. By the dynamics of continuous-time wealth process $\tilde{Z}_t^{\tilde{\pi}}$ in Section 2.3 the dynamics of $\tilde{Z}_t^{\tilde{\pi}^*}$ is derived as

$$\frac{d\tilde{Z}_t^{\tilde{\pi}^*}}{\tilde{Z}_t^{\tilde{\pi}^*}} = \Pi(t, X_t; \theta_{\tilde{\pi}}^*)^\top f_S(X_t; \theta_S^*)dt + \Pi(t, X_t; \theta_{\tilde{\pi}}^*)^\top g_S(X_t; \theta_S^*)^\top dW_t.$$

ASSUMPTION I.4 For each $R > 0$, if $\|x\| \leq R$ and $t \leq T$, we assume that there exists a $C_R > 0$ such that

$$\|\Pi(t, x; \theta_{\tilde{\pi}}^*)\| \vee \|f_S(x; \theta_S^*)\| \vee \|g_S(x; \theta_S^*)\| \vee \|f_X(x; \theta_S^*)\| \vee \|g_X(x; \theta_S^*)\| \leq C_R,$$

and $\Pi(t, x; \theta_\pi^*)$ is locally Lipschitz with Lipschitz constant $C_R$.

For some $p > 2$, there exists a constant $A$ such that

$$\mathbb{E}\big[\sup_{0 \leq t \leq T} |\tilde{Z}_t^{\tilde{\pi}^*}|^p\big] \vee \mathbb{E}\big[\sup_{0 \leq t \leq T} |\tilde{Z}_t^{\lfloor\tilde{\pi}^*\rfloor}|^p\big] \vee \mathbb{E}\big[\sup_{0 \leq t \leq T} \|X_t\|^p\big] \vee \mathbb{E}\big[\sup_{0 \leq t \leq T} \|\log(S_t)\|^p\big] \leq A.$$

Note that Assumption I.4 requires the stochastic processes to have bounded high-order moments. For a specific model like Kim–Omberg, this is not guaranteed to hold for every initial value and SDE coefficient, but one can derive model-specific sufficient conditions for Assumption I.4. In practice when implementing the method, we calculate the empirical moments of wealth, factors and asset prices to approximately check whether Assumption I.4 holds.

With Assumption I.4, we define a stopping time:

DEFINITION I.5 For $R > 0$, define a stopping time

$$\tau_R := \left\{\inf_{0 \leq t \leq T} \big| |\tilde{Z}_t^{\tilde{\pi}^*}| \geq R \text{ or } \|X_t\| \geq R \text{ or } \|\log(S_t)\| \geq R \text{ or } |\tilde{Z}_t^{\lfloor\tilde{\pi}^*\rfloor}| \geq R\right\}.$$

ASSUMPTION I.6 The utility function $U(z)$ has a linear bound on $\mathcal{Z}$:

$$|U(z)| \leq C_U(|z| + 1).$$

Note that Assumption I.6 holds as long as the utility function is bounded at the smallest value in $\mathcal{Z}$. Specifically for power utility, Assumption I.6 is equivalent to setting $\gamma \in (0, 1)$.

ASSUMPTION I.7 There exists $\Delta t' > 0$ such that for any $\Delta t < \Delta t'$, $\lfloor\tilde{\pi}^*\rfloor$ is also an admissible policy.

## I.2 LEMMAS

LEMMA I.8 Consider $n$ non-negative constants $c_1, c_2, \cdots, c_n$. The following inequality is true:

$$(\sum_{i=1}^n c_i)^2 \leq n \sum_{i=1}^n c_i^2.$$

*Proof.* The proof follows the Cauchy-Schwartz inequality. □

LEMMA I.9 $(\theta_\phi^*, \theta_{\tilde{\pi}}^*, \theta_S^*)$ maximizes both $\tilde{V}(\theta_\phi, \theta_{\tilde{\pi}})$ and $L(\theta_\phi, \theta_S)$.

*Proof.* First of all, since $\theta_{\tilde{\pi}^*}$ is defined to be the optimal parameter for the continuous-time policy and $\theta_\phi^*$ is defined to be the true data generating parameter, $(\theta_\phi^*, \theta_\pi^*)$ maximizes $\tilde{V}$. Then, by (23), $(\theta_\phi^*, \theta_S^*)$ maximize $L$. □

LEMMA I.10 For any $\delta > 0$,

$$\mathbb{E}\big[\sup_{0 \leq t \leq T}(\tilde{Z}_t^{\lfloor\tilde{\pi}^*\rfloor} - \tilde{Z}_t^{\tilde{\pi}^*})^2\big] \leq \mathbb{E}\big[\sup_{0 \leq t \leq T}(\tilde{Z}_{t \wedge \tau_R}^{\lfloor\tilde{\pi}^*\rfloor} - \tilde{Z}_{t \wedge \tau_R}^{\tilde{\pi}^*})^2\big] + \frac{2^{p+1}\delta A}{p} + \frac{(p-2)8A}{p\delta^{2/(p-2)}R^p}.$$

*Proof.* Proof by applying Young's inequality. See derivation in Higham et al. (2002, Equation (2.8)). □

LEMMA I.11 For any $t \leq \tau_R$, the difference between the coefficients of the dynamics of $\tilde{Z}_t^{\tilde{\pi}^*}$ and $\tilde{Z}_t^{\lfloor \tilde{\pi}^* \rfloor}$ are bounded by:

$$\left| \sum_{i=1}^{d_S} \Pi(t, X_t; \theta_\pi^*) f_S^i(X_t; \theta_S^*) \tilde{Z}_t^{\tilde{\pi}^*} - \sum_{i=1}^{d_S} \Pi(\lfloor t \rfloor, X_{\lfloor t \rfloor}; \theta_\pi^*) f_S^i(X_t; \theta_S^*) \tilde{Z}_{\lfloor t \rfloor}^{\lfloor \tilde{\pi}^* \rfloor} \frac{S_t^i}{S_{\lfloor t \rfloor}^i} \right|^2$$
$$\leq 5 d_S^2 \exp(2R) C_R^4 \big[ \exp(2R) R^2 |t - \lfloor t \rfloor|^2 + \exp(2R) R^2 |X_t - X_{\lfloor t \rfloor}|^2$$
$$+ R^2 \left\| S_{\lfloor t \rfloor} - S_t \right\|^2 + \exp(2R) \left| \tilde{Z}_t^{\tilde{\pi}^*} - \tilde{Z}_t^{\lfloor \tilde{\pi}^* \rfloor} \right|^2 + \exp(2R) \left| \tilde{Z}_t^{\lfloor \tilde{\pi}^* \rfloor} - \tilde{Z}_{\lfloor t \rfloor}^{\lfloor \tilde{\pi}^* \rfloor} \right|^2 \big],$$

and

$$\left| \sum_{i=1}^{d_S} \left[ \Pi(\lfloor t \rfloor, X_{\lfloor t \rfloor}; \theta_\pi^*) \sum_{j=1}^{d_W} \left( g_S^{ij}(X_t; \theta_S^*) \frac{S_t^i}{S_{\lfloor t \rfloor}^i} \tilde{Z}_{\lfloor t \rfloor}^{\lfloor \tilde{\pi}^* \rfloor} \right) \right] - \sum_{i=1}^{d_S} \left[ \Pi(t, X_t; \theta_\pi^*) \sum_{j=1}^{d_W} \left( g_S^{ij}(X_t; \theta_S^*) \tilde{Z}_t^{\tilde{\pi}^*} \right) \right] \right|^2$$
$$\leq 5 d_S^2 \exp(2R) C_R^4 \bigg[ \exp(2R) R^2 |t - \lfloor t \rfloor|^2 + \exp(2R) R^2 |X_t - X_{\lfloor t \rfloor}|^2$$
$$+ R^2 \left\| S_{\lfloor t \rfloor} - S_t \right\|^2 + \exp(2R) \left| \tilde{Z}_t^{\tilde{\pi}^*} - \tilde{Z}_t^{\lfloor \tilde{\pi}^* \rfloor} \right|^2 + \exp(2R) \left| \tilde{Z}_t^{\lfloor \tilde{\pi}^* \rfloor} - \tilde{Z}_{\lfloor t \rfloor}^{\lfloor \tilde{\pi}^* \rfloor} \right|^2 \bigg].$$

*Proof.*

By triangle inequality

$$\left| \sum_{i=1}^{d_S} \tilde{\pi}_t^i f_S^i(X_t; \theta_S^*) \tilde{Z}_t^{\tilde{\pi}^*} - \sum_{i=1}^{d_S} \tilde{\pi}_{\lfloor t \rfloor}^i f_S^i(X_t; \theta_S^*) \tilde{Z}_{\lfloor t \rfloor}^{\lfloor \tilde{\pi}^* \rfloor} \frac{S_t^i}{S_{\lfloor t \rfloor}^i} \right|$$
$$\leq \sum_{i=1}^{d_S} \frac{1}{S_{\lfloor t \rfloor}^i} |f_S^i(X_t; \theta_S^*)| \big( \left| \tilde{\pi}_t^i \tilde{Z}_t^{\tilde{\pi}^*} S_{\lfloor t \rfloor}^i - \tilde{\pi}_{\lfloor t \rfloor}^i \tilde{Z}_t^{\tilde{\pi}^*} S_{\lfloor t \rfloor}^i \right|$$
$$+ \left| \tilde{\pi}_{\lfloor t \rfloor}^i \tilde{Z}_t^{\tilde{\pi}^*} S_{\lfloor t \rfloor}^i - \tilde{\pi}_{\lfloor t \rfloor}^i \tilde{Z}_{\lfloor t \rfloor}^{\lfloor \tilde{\pi}^* \rfloor} S_{\lfloor t \rfloor}^i \right| + \left| \tilde{\pi}_{\lfloor t \rfloor}^i \tilde{Z}_{\lfloor t \rfloor}^{\lfloor \tilde{\pi}^* \rfloor} S_{\lfloor t \rfloor}^i - \tilde{\pi}_{\lfloor t \rfloor}^i \tilde{Z}_{\lfloor t \rfloor}^{\lfloor \tilde{\pi}^* \rfloor} S_t^i \right| \big).$$

For any $t \leq \tau_R$, we can further bound the right hand side by Assumption I.4

$$\left| \sum_{i=1}^{d_S} \tilde{\pi}_t^i f_S^i(X_t; \theta_S^*) \tilde{Z}_t^{\tilde{\pi}^*} - \sum_{i=1}^{d_S} \tilde{\pi}_{\lfloor t \rfloor}^i f_S^i(X_t; \theta_S^*) \tilde{Z}_{\lfloor t \rfloor}^{\lfloor \tilde{\pi}^* \rfloor} \frac{S_t^i}{S_{\lfloor t \rfloor}^i} \right|$$
$$\leq d_S \exp(R) C_R^2 \big[ \exp(R) R \big( |t - \lfloor t \rfloor| + \left\| X_t - X_{\lfloor t \rfloor} \right\| \big)$$
$$+ \exp(R) \big( \left| \tilde{Z}_t^{\tilde{\pi}^*} - \tilde{Z}_t^{\lfloor \tilde{\pi}^* \rfloor} \right| + \left| \tilde{Z}_t^{\lfloor \tilde{\pi}^* \rfloor} - \tilde{Z}_{\lfloor t \rfloor}^{\lfloor \tilde{\pi}^* \rfloor} \right| \big) + R \left\| S_{\lfloor t \rfloor} - S_t \right\| \big].$$

Then, by Lemma I.8

$$\left| \sum_{i=1}^{d_S} \tilde{\pi}_t^i f_S^i(X_t; \theta_S^*) \tilde{Z}_t^{\tilde{\pi}^*} - \sum_{i=1}^{d_S} \tilde{\pi}_{\lfloor t \rfloor}^i f_S^i(X_t; \theta_S^*) \tilde{Z}_{\lfloor t \rfloor}^{\lfloor \tilde{\pi}^* \rfloor} \frac{S_t^i}{S_{\lfloor t \rfloor}^i} \right|^2$$
$$\leq 5 d_S^2 \exp(2R) C_R^4 \big[ \exp(2R) R^2 \big( |t - \lfloor t \rfloor|^2 + |X_t - X_{\lfloor t \rfloor}|^2 \big)$$
$$+ \exp(2R) \big( \left| \tilde{Z}_t^{\tilde{\pi}^*} - \tilde{Z}_t^{\lfloor \tilde{\pi}^* \rfloor} \right|^2 + \left| \tilde{Z}_t^{\lfloor \tilde{\pi}^* \rfloor} - \tilde{Z}_{\lfloor t \rfloor}^{\lfloor \tilde{\pi}^* \rfloor} \right|^2 \big) + R^2 \left\| S_{\lfloor t \rfloor} - S_t \right\|^2 \big]$$
$$= 5 d_S^2 \exp(2R) C_R^4 \big[ \exp(2R) R^2 |t - \lfloor t \rfloor|^2 + \exp(2R) R^2 |X_t - X_{\lfloor t \rfloor}|^2$$
$$+ R^2 \left\| S_{\lfloor t \rfloor} - S_t \right\|^2 + \exp(2R) \left| \tilde{Z}_t^{\tilde{\pi}^*} - \tilde{Z}_t^{\lfloor \tilde{\pi}^* \rfloor} \right|^2 + \exp(2R) \left| \tilde{Z}_t^{\lfloor \tilde{\pi}^* \rfloor} - \tilde{Z}_{\lfloor t \rfloor}^{\lfloor \tilde{\pi}^* \rfloor} \right|^2 \big].$$

Similarly

$$\left| \sum_{i=1}^{d_S} \left[ \tilde{\pi}_{\lfloor t \rfloor}^i \sum_{j=1}^{d_W} \left( g_S^{ij}(X_t; \theta_S^*) \frac{S_t^i}{S_{\lfloor t \rfloor}^i} \tilde{Z}_{\lfloor t \rfloor}^{\lfloor \tilde{\pi}^* \rfloor} \right) \right] - \sum_{i=1}^{d_S} \left[ \tilde{\pi}_t^i \sum_{j=1}^{d_W} \left( g_S^{ij}(X_t; \theta_S^*) \tilde{Z}_t^{\tilde{\pi}^*} \right) \right] \right|^2$$

$$\leq 5 d_S^2 \exp(2R) C_R^4 \big[ \exp(2R) R^2 |t - \lfloor t \rfloor|^2 + \exp(2R) R^2 \left| X_t - X_{\lfloor t \rfloor} \right|^2$$

$$+ R^2 \left\| S_{\lfloor t \rfloor} - S_t \right\|^2 + \exp(2R) \left| \tilde{Z}_t^{\tilde{\pi}^*} - \tilde{Z}_t^{\lfloor \tilde{\pi}^* \rfloor} \right|^2 + \exp(2R) \left| \tilde{Z}_t^{\lfloor \tilde{\pi}^* \rfloor} - \tilde{Z}_{\lfloor t \rfloor}^{\lfloor \tilde{\pi}^* \rfloor} \right|^2 \big].$$

$\square$

LEMMA I.12  With $\tau_R$ defined in Definition I.5,

$$\mathbb{E} \left\| X_{t \wedge \tau_R} - X_{\lfloor t \wedge \tau_R \rfloor} \right\|^2 \leq 2 C_R^2 (\Delta t^2 + \Delta t),$$

$$\mathbb{E} \left\| S_{t \wedge \tau_R} - S_{\lfloor t \wedge \tau_R \rfloor} \right\|^2 \leq 2 \exp(2R) C_R^2 (\Delta t^2 + \Delta t),$$

$$\mathbb{E} \left\| \tilde{Z}_{\lfloor t \wedge \tau_R \rfloor}^{\lfloor \tilde{\pi}^* \rfloor} - \tilde{Z}_{t \wedge \tau_R}^{\lfloor \tilde{\pi}^* \rfloor} \right\|^2 \leq 2 R^2 \exp(4R) C_R^4 (\Delta t^2 + \Delta t).$$

*Proof.*  By the dynamics of $X_t$ and Lemma I.8, we can derive

$$\mathbb{E} \left\| X_{t \wedge \tau_R} - X_{\lfloor t \wedge \tau_R \rfloor} \right\|^2 \leq 2 \Delta t \mathbb{E} \int_{m_{t \wedge \tau_R} \Delta t}^{t \wedge \tau_R} \| f_X(X_s; \theta_S^*) \|^2 \, ds + 2 \mathbb{E} \left\| \int_{m_{t \wedge \tau_R} \Delta t}^{t \wedge \tau_R} g_X(X_s; \theta_S^*)^\top dW_s \right\|^2,$$

where $m_{t \wedge \tau_R}$ satisfies $m_{t \wedge \tau_R} \Delta t \leq (t \wedge \tau_R) < (m_{t \wedge \tau_R} + 1) \Delta t$. Further, we apply Itô's isometry with stopping time (Lemma D.2), and derive

$$\mathbb{E} \left\| X_{t \wedge \tau_R} - X_{\lfloor t \wedge \tau_R \rfloor} \right\|^2 \leq 2 \Delta t \mathbb{E} \int_{m_{t \wedge \tau_R} \Delta t}^{t \wedge \tau_R} \| f_X(X_s; \theta_S^*) \|^2 \, ds + 2 \mathbb{E} \int_{m_{t \wedge \tau_R} \Delta t}^{t \wedge \tau_R} \| g_X(X_s; \theta_S^*) \|^2 \, ds.$$

By Assumption I.4, we derive

$$\mathbb{E} \left\| X_{t \wedge \tau_R} - X_{\lfloor t \wedge \tau_R \rfloor} \right\|^2 \leq 2 C_R^2 (\Delta t^2 + \Delta t).$$

Similarly

$$\mathbb{E} \left\| S_{t \wedge \tau_R} - S_{\lfloor t \wedge \tau_R \rfloor} \right\|^2 \leq 2 \exp(2R) C_R^2 (\Delta t^2 + \Delta t),$$

and

$$\mathbb{E} \left\| \tilde{Z}_{\lfloor t \wedge \tau_R \rfloor}^{\lfloor \tilde{\pi}^* \rfloor} - \tilde{Z}_{t \wedge \tau_R}^{\lfloor \tilde{\pi}^* \rfloor} \right\|^2 \leq 2 R^2 \exp(4R) C_R^4 (\Delta t^2 + \Delta t).$$

$\square$

LEMMA I.13

$$\mathbb{E} \big[ \sup_{0 \leq t \leq \tau} (\tilde{Z}_{t \wedge t_k}^{\tilde{\pi}^*} - \tilde{Z}_{t \wedge t_k}^{\lfloor \tilde{\pi}^* \rfloor})^2 \big] \leq 10 (T + 4) T d_S^2 C_R^4 \exp(4R) R^2 \big[ \Delta t^2 + 2 C_R^2 (\Delta t^2 + \Delta t)$$

$$+ 2 C_R^2 (\Delta t^2 + \Delta t) + 2 \exp(4R) C_R^4 (\Delta t^2 + \Delta t) \big]$$

$$+ 10 (T + 4) d_S^2 C_R^4 \exp(4R) \int_0^\tau \mathbb{E} \sup_{0 \leq r \leq s} \left| \tilde{Z}_{r \wedge \tau_R}^{\tilde{\pi}^*} - \tilde{Z}_{r \wedge \tau_R}^{\lfloor \tilde{\pi}^* \rfloor} \right|^2 \, ds.$$

*Proof.*

By Cauchy–Schwarz inequality and the dynamics of $\tilde{Z}_t^{\tilde{\pi}^*}$ and $\tilde{Z}_t^{\lfloor \tilde{\pi}^* \rfloor}$, for any $\tau \leq T$

$$\mathbb{E} \big[ \sup_{0 \leq t \leq \tau} (\tilde{Z}_{t \wedge t_k}^{\tilde{\pi}^*} - \tilde{Z}_{t \wedge t_k}^{\lfloor \tilde{\pi}^* \rfloor})^2 \big]$$

$$\leq 2 T \mathbb{E} \left[ \sup_{0 \leq t \leq \tau} \int_0^{t \wedge \tau_R} \left| \sum_{i=1}^{d_S} \tilde{\pi}_s^i f_S^i(X_s; \theta_S^*) \tilde{Z}_s^{\tilde{\pi}^*} - \sum_{i=1}^{d_S} \tilde{\pi}_{\lfloor s \rfloor}^i f_S^i(X_s; \theta_S^*) \tilde{Z}_{\lfloor s \rfloor}^{\lfloor \tilde{\pi}^* \rfloor} \frac{S_s^i}{S_{\lfloor t \rfloor}^i} \right|^2 \, ds \right]$$

$$+ 2 \mathbb{E} \left[ \sup_{0 \leq t \leq \tau} \left| \sum_{j=1}^{d_W} \int_0^{t \wedge \tau_R} \left[ \sum_{i=1}^{d_S} \tilde{\pi}_s^i g^{ij}(X_s; \theta_S^*) \tilde{Z}_s^{\tilde{\pi}^*} - \tilde{\pi}_{\lfloor s \rfloor}^i g^{ij}(X_{\lfloor s \rfloor}; \theta_S^*) S_s^i / S_{\lfloor s \rfloor}^i \right] dW_s^j \right|^2 \right].$$

Then, by Doob's martingale inequality

$$
\mathbb{E}\big[\sup_{0\leq t\leq\tau}(\tilde{Z}^{\tilde{\pi}^*}_{t\wedge t_k}-\tilde{Z}^{\lfloor\tilde{\pi}^*\rfloor}_{t\wedge t_k})^2\big]
$$

$$
\leq 2T\mathbb{E}\bigg[\sup_{0\leq t\leq\tau}\int_0^{t\wedge\tau_R}\bigg|\sum_{i=1}^{d_S}\tilde{\pi}^i_s f^i_S(X_s;\theta^*_S)\tilde{Z}^{\tilde{\pi}^*}_s - \sum_{i=1}^{d_S}\tilde{\pi}^i_{\lfloor s\rfloor}f^i_S(X_s;\theta^*_S)\tilde{Z}^{\lfloor\tilde{\pi}^*\rfloor}_{\lfloor s\rfloor}\frac{S^i_s}{S^i_{\lfloor t\rfloor}}\bigg|^2 ds\bigg]
$$

$$
+8T\mathbb{E}\bigg[\bigg(\sum_{j=1}^{d_W}\int_0^{t\wedge\tau_R}\bigg[\sum_{i=1}^{d_S}\tilde{\pi}^i_s g^{ij}(X_s;\theta^*_S)\tilde{Z}^{\tilde{\pi}^*}_s - \tilde{\pi}^i_{\lfloor s\rfloor}g^{ij}(X_{\lfloor s\rfloor};\theta^*_S)S^i_s/S^i_{\lfloor s\rfloor}\bigg]dW^j_s\bigg)^2\bigg].
$$

Next, we apply Lemma I.11,

$$
\mathbb{E}\big[\sup_{0\leq t\leq\tau}(\tilde{Z}^{\tilde{\pi}^*}_{t\wedge t_k}-\tilde{Z}^{\lfloor\tilde{\pi}^*\rfloor}_{t\wedge t_k})^2\big]
$$

$$
\leq 10(T+4)d_S^2 C_R^4\exp(2R)\mathbb{E}\bigg[\int_0^{\tau\wedge\tau_R}\big(\exp(2R)R^2|s-\lfloor s\rfloor|^2 + \exp(2R)R^2\big|X_s-X_{\lfloor s\rfloor}\big|^2
$$

$$
+R^2\big\|S_{\lfloor s\rfloor}-S_s\big\|^2 + \exp(2R)\big|\tilde{Z}^{\tilde{\pi}^*}_s-\tilde{Z}^{\lfloor\tilde{\pi}^*\rfloor}_s\big|^2 + \exp(2R)\big|\tilde{Z}^{\lfloor\tilde{\pi}^*\rfloor}_s-\tilde{Z}^{\lfloor\tilde{\pi}^*\rfloor}_{\lfloor s\rfloor}\big|^2\big)ds\bigg].
$$

Therefore, combined with I.12,

$$
\mathbb{E}\big[\sup_{0\leq t\leq\tau}(\tilde{Z}^{\tilde{\pi}^*}_{t\wedge t_k}-\tilde{Z}^{\lfloor\tilde{\pi}^*\rfloor}_{t\wedge t_k})^2\big] \leq 10(T+4)Td_S^2 C_R^4\exp(4R)R^2\big[\Delta t^2 + 2C_R^2(\Delta t^2+\Delta t)
$$

$$
+2C_R^2(\Delta t^2+\Delta t)+2\exp(4R)C_R^4(\Delta t^2+\Delta t)\big]
$$

$$
+10(T+4)d_S^2 C_R^4\exp(4R)\int_0^\tau\mathbb{E}\sup_{0\leq r\leq s}\big|\tilde{Z}^{\tilde{\pi}^*}_{r\wedge\tau_R}-\tilde{Z}^{\lfloor\tilde{\pi}^*\rfloor}_{r\wedge\tau_R}\big|^2 ds.
$$

□

LEMMA I.14  With the definitions and assumptions in Section I.1,

$$
\lim_{\Delta t\to 0}\mathbb{E}[(\tilde{Z}^{\lfloor\tilde{\pi}^*\rfloor}_T-\tilde{Z}^{\tilde{\pi}^*}_T)^2]=0.
$$

*Proof.*

By Lemma I.13, we apply the Gronwall inequality and obtain

$$
\mathbb{E}\big[\sup_{0\leq t\leq T}(\tilde{Z}^{\tilde{\pi}^*}_{t\wedge t_k}-\tilde{Z}^{\tilde{\pi}^*}_{t\wedge t_k})^2\big]
$$

$$
\leq 10(T+4)Td_S^2 C_R^4\exp(4R)R^2\big[\Delta t^2 + 2C_R^2(\Delta t^2+\Delta t)
$$

$$
+2C_R^2(\Delta t^2+\Delta t)+2\exp(4R)C_R^4(\Delta t^2+\Delta t)\big]\exp(10(T+4)d_S^2 C_R^4\exp(4R)).
$$

Then, combined with Lemma I.10, for any $\delta>0$,

$$
\mathbb{E}[\sup_{0\leq t\leq T}(\tilde{Z}^{\tilde{\pi}^*}_t-\tilde{Z}^{\lfloor\tilde{\pi}^*\rfloor}_t)^2]
$$

$$
\leq 10(T+4)Td_S^2 C_R^4\exp(4R)R^2\big[\Delta t^2 + 2C_R^2(\Delta t^2+\Delta t)
$$

$$
+2C_R^2(\Delta t^2+\Delta t)+2\exp(4R)C_R^4(\Delta t^2+\Delta t)\big]\exp(10(T+4)d_S^2 C_R^4\exp(4R))
$$

$$
+\frac{2^{p+1}\delta A}{p}+\frac{(p-2)8A}{p\delta^{2/(p-2)}R^p}.
$$

Therefore, $\mathbb{E}[\sup_{0\leq t\leq T}(\tilde{Z}^{\tilde{\pi}^*}_t-\tilde{Z}^{\lfloor\tilde{\pi}^*\rfloor}_t)^2]$ converges to 0 as $\Delta t$ goes to 0.  □

### I.3 PROOF

For ease of presentation, we define

$$\theta := (\theta_\phi, \theta_\pi, \theta_S), \theta^* := (\theta_\phi^*, \theta_{\tilde{\pi}}^*, \theta_S^*) \text{ and } \theta_{\Delta t}^* := (\theta_{\phi, \Delta t}^*, \theta_{\pi, \Delta t}^*, \theta_{S, \Delta t}^*).$$

Note that every discrete-time admissible policy is a continuous-time admissible policy. Thus, the continuous-time admissible policy set includes the discrete-time admissible policy set. Therefore,

$$\tilde{V}(\theta^*) \geq V_{\Delta t}^*.$$

In other words, it is enough to bound $\tilde{V}(\theta^*) - V(\theta_{\Delta t}^*)$ for the proof. By Lemma I.9, $\theta^*$ maximizes $\tilde{V}$ and $L$ simultaneously, leading to

$$\tilde{V}(\theta^*) - V(\theta_{\Delta t}^*) \leq \frac{\tilde{H}(\theta^*) - H(\theta_{\Delta t}^*)}{1 - \lambda}.$$

By Assumption I.7, for $\Delta t \leq \Delta t'$, $\theta^*$ is also an admissible parameter $\theta^* \in \mathcal{A}$, leading to $H(\theta^*) - H(\theta_{\Delta t}^*) \leq 0$. Further, for any $\delta > 0$, by adding and subtracting equal terms,

$$\tilde{V}(\theta^*) - V(\theta_{\Delta t}^*)$$
$$\leq \frac{1}{1 - \lambda}[\tilde{H}(\theta^*) - H(\theta^*) + H(\theta^*) - H(\theta_{\Delta t}^*)] \tag{24}$$
$$\leq \frac{1}{1 - \lambda}\left|\tilde{H}(\theta^*) - H(\theta^*)\right|.$$

Next, we focus on $\left|\tilde{H}(\theta^*; \delta) - H(\theta^*; \delta)\right|$, which by definition has

$$\left|\tilde{H}(\theta^*; \delta) - H(\theta^*; \delta)\right| = (1 - \lambda)\left|\mathbb{E}[U(\tilde{Z}_T^{\tilde{\pi}^*}; \delta) - U(\tilde{Z}_T^{\lfloor\tilde{\pi}^*\rfloor}; \delta)]\right|,$$

where $\lambda L(\theta^*)$ in both $\tilde{H}(\theta^*; \delta)$ and $H(\theta^*; \delta)$ omit each other. By Lemma I.14, we have $\lim_{\Delta t \to 0} \tilde{Z}_T^{\lfloor\tilde{\pi}^*\rfloor} \xrightarrow{P} \tilde{Z}_T^{\tilde{\pi}^*}$. Since $U(z; \delta)$ is a continuous function, we implement the continuous mapping theorem and derive

$$\lim_{\Delta t \to 0} U(\tilde{Z}_T^{\lfloor\tilde{\pi}^*\rfloor}; \delta) \xrightarrow{P} U(\tilde{Z}_T^{\tilde{\pi}^*}; \delta). \tag{25}$$

By assumption I.4, $\left\{\tilde{Z}_T^{\lfloor\tilde{\pi}^*\rfloor}\right\}_{\Delta t < \Delta t'}$ with different finite $\Delta t$ is uniformly integrable. Then, following Assumption I.6, $U(\tilde{Z}_T^{\lfloor\tilde{\pi}^*\rfloor}; \delta)$ is also uniformly integrable since $U(z; \delta)$ has a linear bound. Combined with (25), we derive

$$\lim_{\Delta t \to 0} \mathbb{E}[U(\tilde{Z}_T^{\lfloor\tilde{\pi}^*\rfloor}; \delta)] \to \mathbb{E}[U(\tilde{Z}_T^{\tilde{\pi}^*}; \delta)],$$

which finishes the proof.

## J EXTENDED RESULTS OF THEOREM 4.2

In this section, we study the non-asymptotic guarantees on the performance of FaLPO.

### J.1 ANOTHER VERSION OF THEOREM 4.2

DEFINITION J.1 For two random vectors $v$ and $w$, we define the trace of the covariance matrix as

$$\text{Var}(v) := \mathbb{E}[\|v\|_2^2 - \|\mathbb{E}[v]\|_2^2],$$
$$\text{Cov}(v, w) := \mathbb{E}[v^\top w] - \mathbb{E}[v]^\top \mathbb{E}[w].$$

Further, we use $\text{Var}_\theta(v)$ and $\text{Cov}_\theta(v, w)$ to denote the conditional version of the two given $\theta$:

$$\text{Var}_\theta(v) := \mathbb{E}[\|v\|_2^2 - \|\mathbb{E}[v|\theta]\|_2^2 |\theta],$$
$$\text{Cov}_\theta(v, w) := \mathbb{E}[v^\top w|\theta] - \mathbb{E}[v|\theta]^\top \mathbb{E}[w|\theta].$$

Note that it is challenging to theoretically analyze a non-convex stochastic optimization (5), while there are various ad-hoc procedures providing good empirical performances. To provide theoretical analysis, in this section, we study a projection-based version of FaLPO (Algorithm 2). Specifically, the learning/optimization process is conducted in a bounded parameter space $\mathcal{B}$, under which we assume that the objective function is strongly concave regarding the parameters.

---

**Algorithm 2** Projected FaLPO

---

1: **Input:** Hyperparameter $\lambda$, learning rate $\eta$, number of iterations $N$, the strongly concave region $\mathcal{B}$, and batch size $B$.
2: **Output:** $\theta_\phi$, $\theta_\pi$, and $\theta_S$
3: Initialize neural networks with initial parameters $(\theta_\phi, \theta_\pi, \theta_S) \in \mathcal{B}$.
4: Parameterize the policy function by (3).
5: **for** $n \in [N]$ **do**
6:     Collect $B$ trajectories.
7:     Estimate the gradients of $H$ following the procedure in Appendix C with the parameter $\lambda$.
8:     Update $\theta_S$ and $\theta_R$ with learning rate $\eta$ by gradients.
9:     Project the achieved update to $\mathcal{B}$.
10: **end for**
11: **Return** $\theta_\phi$, $\theta_\pi$, and $\theta_S$.

---

For ease of presentation, we define

$$\theta^* := (\theta_\phi^*, \theta_{\tilde{\pi}}^*, \theta_S^*), \theta_{\Delta t}^* := (\theta_{\phi,\Delta t}^*, \theta_{\pi,\Delta t}^*, \theta_{S,\Delta t}^*) \text{ and } \theta^\dagger := (\theta_{\phi,\Delta t}^\dagger, \theta_{\pi,\Delta t}^\dagger, \theta_{S,\Delta t}^\dagger).$$

Let $\theta_n$ be the estimation after the $n^{\text{th}}$ iteration, and $\bar{\theta} := \frac{\sum_{n=0}^{N-1} \theta_n}{N}$ the average estimation. It is a common technique to consider the average estimation $\bar{\theta}$ instead of the final estimations $\theta_N$ for such analysis. Then, we provide a new version of Theorem 4.2.

DEFINITION J.2 With the gradient estimations discussed in Appendix C, we define

$$\tilde{\nabla} H_k(\theta) := (1 - \lambda)\tilde{\nabla} V_k(\theta) + \lambda \tilde{\nabla} L_k(\theta).$$

THEOREM J.3 With assumptions in Section J.2, $\lambda \in [0, 1)$, and $\eta < \frac{1}{C_L}$ as the learning rate,

$$\mathbb{E}[V_{\Delta t}^* - V(\bar{\theta})]$$

$$\leq \frac{\tilde{H}(\theta^*) - H(\theta_{\Delta t}^*)}{1 - \lambda} + \frac{H(\theta_{\Delta t}^*) - H(\theta^\dagger)}{1 - \lambda} + \frac{C_\mathcal{B} \log(N)}{2N(1 - \lambda)}$$

$$+ \frac{1}{2NB(1 - \lambda)} \mathbb{E}\Bigg[ \sum_{n=0}^{N-1} \frac{1}{n + 1} [(1 - \lambda)^2 \text{Var}_{\theta_n} \left( \tilde{\nabla} V_k(\theta_n) \right) + \lambda^2 \text{Var}_{\theta_n} \left( \tilde{\nabla} L_k(\theta_n) \right)$$

$$+ 2\lambda(1 - \lambda)\text{Cov}_{\theta_n}(\tilde{\nabla} V_k(\theta_n), \tilde{\nabla} L_k(\theta_n))] \Bigg].$$

Also, there exits situations where a $\lambda \in (0, 1)$ provides smaller value for $(1 - \lambda)^2 \text{Var} \left( \tilde{\nabla} H_k(\theta_n) \right) + \lambda^2 \text{Var} \left( \tilde{\nabla} L_k(\theta_n) \right) + 2\lambda(1 - \lambda)\text{Cov}(\tilde{\nabla} H_k(\theta_n), \tilde{\nabla} R_k(\theta_n))$ than $\lambda = 0$. In other words, there exist cases where tuning $\lambda$ may provide better performances.

## J.2 ASSUMPTIONS

ASSUMPTION J.4 There exits a constant $C_\mathcal{B} > 0$ such that the parameter region $\mathcal{B}$ is a convex set and satisfies the following conditions

1. In $\mathcal{B} \subseteq \mathcal{A}$, $H(\theta_{\phi,\Delta t}, \theta_{\pi,\Delta t}, \theta_{S,\Delta t})$ is locally $m$-strongly concave with a local maximal point $(\theta_{\phi,\Delta t}^\dagger, \theta_{P,\Delta t}^\dagger, \theta_{S,\Delta t}^\dagger) \in \mathcal{B}$.

2. For any $\theta \in \mathcal{B}$, $\|\theta\| \leq C_\mathcal{B}$ .

3. For any $\theta \in \mathcal{B}$, the expectation of the gradient estimation is bounded by

$$\left\| \mathbb{E}\left[ \frac{\sum_{k=1}^{B} \tilde{\nabla} H_k(\theta)}{B} \right] \right\|^2 \leq C_{\mathcal{B}}.$$

These assumptions are widely used in existing analysis (Papini et al., 2018; Karimi et al., 2019; Agarwal et al., 2021; Bhandari & Russo, 2019; Wang et al., 2019; Xu et al., 2020).

ASSUMPTION J.5  At the $n^{\text{th}}$ iteration, we use the learning rate as $\eta_{n+1} = \frac{1}{nm}$.

Note that in practice we will tune the learning rate $\eta$ as a hyperparameter, since we may not know $m$. However, it is a common practice to set the learning rate as in Assumption J.5 (Hazan & Kale, 2011; Nemirovski et al., 2009; Shalev-Shwartz et al., 2011)

### J.3  TECHNICAL LEMMAS FOR THEOREM 4.2

LEMMA J.6  With Assumption J.4 and J.5, we have

$$H(\theta^\dagger) - H(\bar{\theta}) \leq \frac{1}{2N} \sum_{n=0}^{N-1} \left[ \frac{1}{n+1} \left\| \frac{\sum_{k=1}^{B} \tilde{\nabla} H_k(\theta_n)}{B} \right\|^2 \right]$$
$$+ \frac{1}{N} \sum_{n=0}^{N-1} \left[ \left( \nabla H(\theta_n) - \frac{\sum_{k=1}^{B} \tilde{\nabla} H_k(\theta_n)}{B} \right)^\top (\theta^\dagger - \theta_n) \right].$$

*Proof.*  By the strong concavity of $H$ in Assumption J.4,

$$\nabla H(\theta_n)(\theta^\dagger - \theta_n) \geq H(\theta^\dagger) - H(\theta_n) + \frac{m}{2} \left\| \theta_n - \theta^\dagger \right\|^2. \tag{26}$$

Further, since $\theta_{n+1}$ is the projection of $\theta_n + \eta_{n+1} \frac{\sum_{k=1}^{B} \tilde{\nabla} H_k(\theta_n)}{B}$ to $\mathcal{B}$, the projection satisfies

$$\left\| \theta_n + \eta_{n+1} \frac{\sum_{k=1}^{B} \tilde{\nabla} H_k(\theta_n)}{B} - \theta^\dagger \right\|^2 \geq \left\| \theta_{n+1} - \theta^\dagger \right\|^2,$$

which suggests

$$\left\| \theta_n - \theta^\dagger \right\|^2 - \left\| \theta_{n+1} - \theta^\dagger \right\|^2$$
$$\geq \left\| \theta_n - \theta^\dagger \right\|^2 - \left\| \theta_n + \eta_{n+1} \frac{\sum_{k=1}^{B} \tilde{\nabla} H_k(\theta_n)}{B} - \theta^\dagger \right\|^2$$
$$= -\eta_{n+1} \frac{\sum_{k=1}^{B} \tilde{\nabla} H_k(\theta_n)}{B}^\top \left( 2\theta_n + \eta_{n+1} \frac{\sum_{k=1}^{B} \tilde{\nabla} H_k(\theta_n)}{B} - 2\theta^\dagger \right) \tag{27}$$
$$= -\eta_{n+1}^2 \left\| \frac{\sum_{k=1}^{B} \tilde{\nabla} H_k(\theta_n)}{B} \right\|_2^2 - 2\eta_{n+1} \left( \theta_n - \theta^\dagger \right)^\top \frac{\sum_{k=1}^{B} \tilde{\nabla} H_k(\theta_n)}{B}.$$

We reorder (27) and derive

$$-\left( \theta_n - \theta^\dagger \right)^\top \frac{\sum_{k=1}^{B} \tilde{\nabla} H_k(\theta_n)}{B} \leq \frac{1}{2\eta_{n+1}} \left( \left\| \theta_n - \theta^\dagger \right\|^2 \right.$$
$$\left. + \eta_{n+1}^2 \left\| \frac{\sum_{k=1}^{B} \tilde{\nabla} H_k(\theta_n)}{B} \right\|_2^2 - \left\| \theta_{n+1} - \theta^\dagger \right\|^2 \right).$$

Taking the result back to (26):

$$
\begin{aligned}
H(\theta^\dagger) - H(\theta_n) \leq & \left( \nabla H(\theta_n) - \frac{\sum_{k=1}^{B} \tilde{\nabla} H_k(\theta_n)}{B} \right)^\top (\theta^\dagger - \theta_n) \\
& + \frac{1}{2\eta_{n+1}} \left( \left\| \theta_n - \theta^\dagger \right\|^2 - \left\| \theta_{n+1} - \theta^\dagger \right\|^2 + \eta_{n+1}^2 \left\| \frac{\sum_{k=1}^{B} \tilde{\nabla} H_k(\theta_n)}{B} \right\|^2 \right) \\
& - \frac{m}{2} \left\| \theta_n - \theta^\dagger \right\|^2, \\
= & \left( \nabla H(\theta_n) - \frac{\sum_{k=1}^{B} \tilde{\nabla} H_k(\theta_n)}{B} \right)^\top (\theta^\dagger - \theta_n) + \frac{1}{2}(\eta_{n+1}^{-1} - m) \left\| \theta_n - \theta^\dagger \right\|^2 \\
& - \frac{1}{2\eta_{n+1}} \left\| \theta_{n+1} - \theta^\dagger \right\|^2 + \frac{1}{2}\eta_{n+1} \left\| \frac{\sum_{k=1}^{B} \tilde{\nabla} H_k(\theta_n)}{B} \right\|^2.
\end{aligned}
$$

$$(28)$$

By averaging over $n$ with Assumption J.5 we get

$$
\begin{aligned}
H(\theta^\dagger) - H(\bar{\theta}) \leq & \sum_{n=0}^{N-1} (H(\theta^\dagger) - H(\theta_n))/N \\
\leq & \frac{1}{2N} \sum_{n=0}^{N-1} \left[ \frac{1}{n+1} \left\| \frac{\sum_{k=1}^{B} \tilde{\nabla} H_k(\theta_n)}{B} \right\|^2 \right] \\
& + \frac{1}{N} \sum_{n=0}^{N-1} \left[ \left( \nabla H(\theta_n) - \frac{\sum_{k=1}^{B} \tilde{\nabla} H_k(\theta_n)}{B} \right)^\top (\theta^\dagger - \theta_n) \right],
\end{aligned}
$$

where the first inequality is due to condition 1 of Assumption J.4. $\qquad \square$

### J.4 PROOF OF THEOREM 4.2

Note that every discrete-time admissible policy is a continuous-time admissible policy. Thus, the continuous-time admissible policy set includes the discrete-time admissible policy set. Therefore,
$$
\tilde{V}(\theta^*) \geq V_{\Delta t}^*.
$$
Therefore, it is enough to bound $\tilde{V}(\theta^*) - V(\bar{\theta})$ for the proof. By Lemma I.9, $\theta^*$ maximizes $\tilde{V}$ and $L$ simultaneously. Therefore,
$$
\tilde{V}(\theta^*) - V(\bar{\theta}) \leq \frac{\tilde{H}(\theta^*) - H(\bar{\theta})}{1 - \lambda} = \frac{\tilde{H}(\theta^*) - H(\theta_{\Delta t}^*) + H(\theta_{\Delta t}^*) - H(\theta^\dagger) + H(\theta^\dagger) - H(\bar{\theta})}{1 - \lambda}.
$$

$$(29)$$

Then, we use the convergence result for Algorithm 2 detailed by Lemma J.6 :

$$
\begin{aligned}
H(\theta^\dagger) - H(\bar{\theta}) \leq & \frac{1}{2N} \sum_{n=0}^{N-1} \left[ \frac{1}{n+1} \left\| \frac{\sum_{k=1}^{B} \tilde{\nabla} H_k(\theta_n)}{B} \right\|^2 \right] \\
& + \frac{1}{N} \sum_{n=0}^{N-1} \left[ \left( \nabla H(\theta_n) - \frac{\sum_{k=1}^{B} \tilde{\nabla} H_k(\theta_n)}{B} \right)^\top (\theta^\dagger - \theta_n) \right].
\end{aligned}
$$

$$(30)$$

Then, we take expectation on both sides

$$
\begin{aligned}
\mathbb{E}[H(\theta^\dagger) - H(\bar{\theta})] \leq & \frac{1}{2N} \sum_{n=0}^{N-1} \mathbb{E} \left[ \frac{1}{n+1} \left\| \frac{\sum_{k=1}^{B} \tilde{\nabla} H_k(\theta_n)}{B} \right\|^2 \right] \\
& + \frac{1}{N} \sum_{n=0}^{N-1} \mathbb{E} \left[ \left( \nabla H(\theta_n) - \frac{\sum_{k=1}^{B} \tilde{\nabla} H_k(\theta_n)}{B} \right)^\top (\theta^\dagger - \theta_n) \right].
\end{aligned}
$$

$$(31)$$

For the last component of (31), since $\tilde{\nabla} H_k(\theta_n)$ is an unbiased gradient estimator:

$$\mathbb{E}\left[\left(\nabla H(\theta_n) - \frac{\sum_{k=1}^{B} \tilde{\nabla} H_k(\theta_n)}{B}\right)^{\top} (\theta^{\dagger} - \theta_n)\right]$$

$$= \mathbb{E}\left[\mathbb{E}\left[\left(\nabla H(\theta_n) - \frac{\sum_{k=1}^{B} \tilde{\nabla} H_k(\theta_n)}{B}\right)^{\top} (\theta^{\dagger} - \theta_n)\Big|\theta_n\right]\right] = 0.$$

Then, for the first component in (31)

$$\mathbb{E}\left[\frac{1}{n+1}\left\|\frac{\sum_{k=1}^{B} \tilde{\nabla} H_k(\theta_n)}{B}\right\|^2\right]$$

$$= \mathbb{E}\left[\mathbb{E}\left[\frac{1}{n+1}\left\|\frac{\sum_{k=1}^{B} \tilde{\nabla} H_k(\theta_n)}{B}\right\|^2\Big|\theta_n\right]\right]$$

$$= \frac{1}{n+1}\mathbb{E}\left[\mathrm{Var}_{\theta_n}\left(\frac{\sum_{k=1}^{B} \tilde{\nabla} H_k(\theta_n)}{B}\right) + \left\|\mathbb{E}\left[\frac{\sum_{k=1}^{B} \tilde{\nabla} H_k(\theta_n)}{B}\Big|\theta_n\right]\right\|^2\right]$$

$$\leq \frac{1}{n+1}\mathbb{E}\left[\mathrm{Var}_{\theta_n}\left(\frac{\sum_{k=1}^{B} \tilde{\nabla} H_k(\theta_n)}{B}\right) + C_{\mathcal{B}}^2\right],$$

where the last inequality is due to condition 3 of Assumption J.4. Then, (31) can be further derived as

$$\mathbb{E}[H(\theta^{\dagger}) - H(\bar{\theta})] \leq \frac{1}{2N}\sum_{n=0}^{N-1}\mathbb{E}\left[\frac{1}{n+1}\mathrm{Var}_{\theta_n}\left(\frac{\sum_{k=1}^{B} \tilde{\nabla} H_k(\theta_n)}{B}\right) + \frac{1}{n+1}C_{\mathcal{B}}\right].$$

As a result,

$$\mathbb{E}[H(\theta_{\Delta t}^*) - H(\bar{\theta})]$$

$$= \mathbb{E}[H(\theta_{\Delta t}^*) - H(\theta^{\dagger}) + H(\theta^{\dagger}) - H(\bar{\theta})]$$

$$\leq H(\theta_{\Delta t}^*) - H(\theta^{\dagger}) + \frac{C_{\mathcal{B}}\log(N)}{2N}$$

$$+ \mathbb{E}\sum_{n=0}^{N-1}\frac{\eta[(1-\lambda)^2\mathrm{Var}_{\theta_n}\left(\tilde{\nabla} V_k(\theta_n)\right) + \lambda^2\mathrm{Var}_{\theta_n}\left(\tilde{\nabla} L_k(\theta_n)\right)]}{BN(n+1)}$$

$$+ \mathbb{E}\sum_{n=0}^{N-1}\frac{\eta[2\lambda(1-\lambda)\mathrm{Cov}_{\theta_n}(\tilde{\nabla} V_k(\theta_n), \tilde{\nabla} L_k(\theta_n))]}{BN(n+1)}\Bigg\}.$$

Taking the results back to (29) we finish the proof.

## K  EXTENDED RESULTS FOR SYNTHETIC EXPERIMENTS

For synthetic portfolio optimization, we provide details for drift and volatility (Appendix K.1), data generation (Appendix K.2), hyperparameter tuning (Appendix K.3), and extended experimental results (Appendix K.4). We consider 21-day trading, and generate 1000 trajectories with 21 observations for training, 1000 for validation, and 1000 for testing. To compare different methods, we calculate the average terminal utility as the metric.

### K.1  DRIFT AND VOLATILITY

Drift and volatility are two important concepts characterising the strength of signal and noise in financial markets. To demonstrate this, for an asset price $S_t^i$ and time interval $\Delta t$, define return as:

$$return_t^i = \frac{S_{t+\Delta t}^i - S_t^i}{S_t^i}.$$

The $return_t^i$ can be daily, monthly or yearly, depending on the length of $\Delta t$. For a specific asset, drift ($f^i(X_t; \theta_S^*)$ in (2)) is approximately the expectation of the return, while volatility ($g^i(X_t; \theta_S^*)$) is approximately the return's standard deviation. Given multiple assets, drift ($f(X_t; \theta_S^*)$) is a vector and volatility ($g(X_t; \theta_S^*)$) is a matrix. When generating synthetic data (Appendix K.2), we fix the scale of drift and vary the scale of volatility, which is defined as the average value of each component.

## K.2 DATA GENERATION

We simulate data for $S_t$ and $X_t$ following SDE (6). To this end, drift and volatility are randomly picked while mimicking the historical stock price data, with an average annual return around as 0.1 and average annual volatility in $\{0.1, 0.2, 0.3\}$, leading to a daily return around $\frac{0.1}{252}$ and a daily volatility around $\{0.1/252, 0.2/252, 0.3/252\}$. The true representation function is selected as a component-wise exponential operation. Then, we discritize the SDE following the explicit Euler method, and generate data accordingly (Beskos & Roberts, 2005).

The specific configurations for data generation is:

- Define two scalars: $C_d$, and $C_v$ determining the scale of drift and volatility:

$$C_d = 0.1/252, , \text{ and } C_v \in \{0.1/252, 0.2/252, 0.3/252\}.$$

- $\sigma$ is selected as a random matrix, whose components follow a uniform distribution in $[0.5C_v, 1.5C_v]$.

- $v$ is selected as a random matrix, whose components follow a uniform distribution in $[-1.5C_dC_v, 1.5C_dC_v]$.

- $\mu$ is selected as a diagonal matrix whose diagonal components follow a uniform distribution in $[0.9, 1]$.

- The initial values of $X$ are randomly generated from a uniform distribution on $[-2C_d, 2C_d]$.

- The initial prices of assets are randomly generated from a uniform distribution in $[20, 30]$.

Note that the design makes sure that the simulated price has approximately a yearly return of 0.1 and yearly volatility in $\{0.1, 0.2, 0.3\}$. Table 4 reports the experimental setup parameters.

| Experiment Configurations | Values |
|---|---|
| Time Interval $\Delta t$ | 1 (Day) |
| Terminal Time $T$ | 21 |
| Scale of Annual Drift | 0.1 |
| Scale of Annual Volatility | $\{0.1, 0.2, 0.3\}$ |
| Number of Simulated Trajectories | 1000 |
| Utility Function | $\{Power, Exponential\}$ |
| Risk Aversion $\gamma$ | $\{0.1, 3, 5, 10\}$ |
| Number of Replications under Each Hyperparameter | 5 |
| Compute Resources | AWS ec2 m5ad.24xlarge |

Table 4: Setup for synthetic experiments

## K.3 EARLY STOPPING AND HYPERPARAMETER TUNING

For better performance, we conduct early stopping for all methods using the average validation utility with the patience as 5 steps. The considered hyperparameters include the learning rate, $\lambda$, and batch size. For each configuration, we conduct training for 5 times, and average the results. Then, we pick the configuration providing the best average validation utility, and test it on the test data and calculate the average test utility per trajectory. The tuning process is conducted using the software wandb (Biewald, 2020). Table 5 reports the hyperparameter values.

| Hyperparameters | Values |
|---|---|
| Batch Size | $\{100, 50\}$ |
| $\lambda$ | $\{0, 05, 0.1, 0.9\}$ |
| Learning Rate | $\{0.0005, 0.001, 0.01, 0.1\}$ |

Table 5: Hyperparameters for synthetic experiments

## K.4 SYNTHETIC EXPERIMENT RESULTS

To gain a more holistic understanding of the performance of FaLPO in a variety of settings, we conduct experiments under different number of stocks to be traded (Appendix K.4.1), different risk preferences (Appendix K.4.2), and alternative utility functions (Appendix K.4.3). Finally, we also compare the performance of various methods under the Merton model as a sanity check (Appendix K.4.4).

### K.4.1 SYNTHETIC EXPERIMENT RESULTS WITH DIFFERENT DIMENSIONS

Tables 6 and 7 report the synthetic experiment results with the number of simulated stocks $(d_S)$ varying in $\{10, 15\}$. The performance is not strictly negatively correlated with the number of dimensions of the problem or the annual volatility in simulation. The reason is that the noise in the problem is indeed determined by the whole volatility matrix $\sigma$, which is randomly generated in the synthetic experiment (Appendix K.2). In other words, the dimension and average scale cannot fully characterize the extent of the noise in a synthetic task.

| | Annual Volatility in Simulation | 0.1 | 0.2 | 0.3 |
|---|---|---|---|---|
| | **FaLPO** | $-0.465 \pm 0.446$ | $-1.35 \pm 0.155$ | $-2.737 \pm 0.219$ |
| | DDPG | $-1.650 \pm 0.456$ | $-3.30 \pm 1.294$ | $-5.495 \pm 1.269$ |
| | SLAC | $-0.750 \pm 0.210$ | $-5.50 \pm 0.011$ | $-6.160 \pm 0.012$ |
| Methods | RichID | $-3.350 \pm 0.111$ | $-5.65 \pm 0.102$ | $-6.325 \pm 0.048$ |
| | CT-MB-RL | $-2.850 \pm 0.014$ | $-5.35 \pm 0.020$ | $-6.160 \pm 0.026$ |
| | MMMC | $-4.723 \pm 7.619$ | $-5.602 \pm 4.299$ | $-6.124 \pm 3.217$ |

Table 6: Average terminal utility after tuning with standard deviation for synthetic data with $d_S = 10$ and $d_W = 10$.

| | Annual Volatility in Simulation | 0.1 | 0.2 | 0.3 |
|---|---|---|---|---|
| | **FaLPO** | $-2.463 \pm 3.744$ | $-1.021 \pm 0.278$ | $-2.243 \pm 0.547$ |
| | DDPG | $-3.976 \pm 1.428$ | $-1.443 \pm 0.751$ | $-5.205 \pm 1.858$ |
| | SLAC | $-4.749 \pm 0.139$ | $-6.129 \pm 0.016$ | $-6.526 \pm 0.012$ |
| Methods | RichID | $-4.973 \pm 0.448$ | $-6.321 \pm 0.038$ | $-6.641 \pm 0.022$ |
| | CT-MB-RL | $-3.074 \pm 0.014$ | $-5.714 \pm 0.023$ | $-6.363 \pm 0.021$ |
| | MMMC | $-5.388 \pm 5.688$ | $-6.465 \pm 4.978$ | $-7.155 \pm 5.965$ |

Table 7: Average terminal utility after tuning with standard deviation for synthetic data with $d_S = 15$ and $d_W = 15$.

### K.4.2 SYNTHETIC EXPERIMENT RESULTS WITH DIFFERENT VALUES OF $\gamma$

Tables 8 and 9 report experimental results with $d_S = 10$, $d_W = 10$, and $\gamma \in \{3, 10\}$ for an exponential utility. FaLPO outperforms the competing methods in most scenarios.

| Methods | Annual Volatility in Simulation | 0.1 | 0.2 | 0.3 |
|---|---|---|---|---|
| | **FaLPO** | $-0.003 \pm 0.0021$ | $-0.0055 \pm 0.0008$ | $-0.0132 \pm 0.0028$ |
| | DDPG | $-0.003 \pm 0.001$ | $-0.0105 \pm 0.006$ | $-0.0205 \pm 0.0034$ |
| | SLAC | $-0.003 \pm 0.0007$ | $-0.0153 \pm 0.0013$ | $-0.0192 \pm 0.0011$ |
| | RichID | $-0.012 \pm 0.0005$ | $-0.0188 \pm 0.0002$ | $-0.0211 \pm 0.0$ |
| | CT-MB-RL | $-0.01 \pm 0.0$ | $-0.0179 \pm 0.0$ | $-0.0206 \pm 0.0$ |
| | MMMC | $-0.0162 \pm 0.0212$ | $-0.0194 \pm 0.0135$ | $-0.0210 \pm 0.0102$ |

Table 8: Average terminal utility after tuning with standard deviation for synthetic data with $\gamma = 10$.

| Methods | Annual Volatility in Simulation | 0.1 | 0.2 | 0.3 |
|---|---|---|---|---|
| | **FaLPO** | $-4.575 \pm 3.325$ | $-17.9358 \pm 2.3349$ | $-56.0405 \pm 13.0502$ |
| | DDPG | $-23.113 \pm 4.3472$ | $-51.6559 \pm 11.7981$ | $-50.0399 \pm 13.8451$ |
| | SLAC | $-23.514 \pm 11.7077$ | $-68.6816 \pm 0.0254$ | $-76.1371 \pm 0.0355$ |
| | RichID | $-44.629 \pm 1.8797$ | $-69.481 \pm 0.9413$ | $-77.232 \pm 0.1091$ |
| | CT-MB-RL | $-34.842 \pm 0.4686$ | $-65.41 \pm 0.1331$ | $-75.4364 \pm 0.122$ |
| | MMMC | $-59.2338 \pm 77.4511$ | $-70.8667 \pm 49.2500$ | $-76.8619 \pm 37.448$ |

Table 9: Average terminal utility after tuning with standard deviation for synthetic data with $\gamma = 3$.

### K.4.3 SYNTHETIC EXPERIMENT RESULTS WITH POWER UTILITY

We also conduct synthetic experiments maximizing the expected power utility for portfolio optimization. The results are summarized in Figure 5.

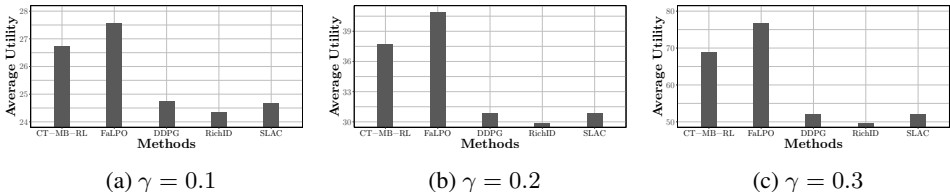

(a) $\gamma = 0.1$         (b) $\gamma = 0.2$         (c) $\gamma = 0.3$

Figure 5: Average Terminal Power Utility

### K.4.4 SYNTHETIC EXPERIMENT FOR THE MERTON CASE

As a sanity check, we study a Merton problem where the optimal performance can be mathematically derived, in order to compare the performance of FaLPO to the optimal one. We simulate data following a Merton model in Appendix F.1, with $d_S = 10$, $d_W = 10$. With an exponential utility function with $\gamma = 5$, according to Lemma F.1, the optimal policy can be derived as

$$\tilde{\pi}^* = \frac{\mu(\sigma\sigma^\top)^{-1}}{Z_t}. \tag{32}$$

Further, by taking (32) back into (14), we can derive the optimal expected terminal utility as

$$\max_{\tilde{\pi}_t} \mathbb{E}_{\tilde{\pi}}[U(Z_T^{\tilde{\pi}})|z_0] = -\frac{e^{-\gamma z_0}}{\gamma} e^{-\frac{1}{2}\mu^\top(\sigma\sigma^\top)^{-1}\mu T},$$

which is the theoretically optimal performance. Then, to implement FaLPO, we generate fake features which are independent from the asset prices: the optimal policy is not dependent on these features. Ideally, FaLPO should be able to automatically ignore the fake features, and deliver performance similar to the theoretically optimal derivation. The results of FaLPO, MMMC, and the theoretically optimal derivation are reported in Figure 6. Note that FaLPO achieves slightly worse performance compared to the other two. The reason for the slight suboptimality of FaLPO in the Merton case is twofold: i. the expected terminal utility is derived for a continuous-time policy while FaLPO learns a

discrete-time policy with time interval $\Delta t$; ii. FaLPO uses an over-complicated model with stochastic factors, while the true data generating process follows a Merton model without stochastic factors.

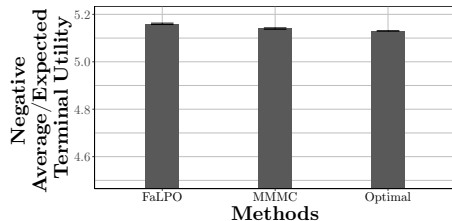

Figure 6: Negative average terminal utility of FaLPO and MMMC, and negative expected optimal terminal utility. The smaller the better.

# L    EXTENDED RESULTS OF REAL-WORLD STOCK TRADING

## L.1    PROTOCOL

We consider 21-day stock trading in four different stock sectors using the daily stock price data from Yahoo finance between January 4, 2006 and April 1, 2022. More specifically, we use the adjusted close price as the daily trading price. For factors, we consider economic indexes, technical analysis indexes (generated by python package TA), and sector-specific features such as oil prices, gold prices, and related ETF prices, leading to around 30 factors for each sector. In each sector we select 10 stocks according to the availability and trading volume in the considered time range. The considered sectors, stocks, and the features are provided in Table 10. We consider the same competing methods in Section 5.1 and compare the performance using the average achieved terminal utility over different trajectories as the metric. The larger the utility the better.

| Sectors | Stocks | Features for Factors |
|---|---|---|
| Energy | APA, COP, CVX, HAL, HES, MRO, OKE, OXY, VLO, WMB | SP500 returns, MACD of stock prices, RSI of stock prices, oil prices, gasoline prices US Dollar/USDX - Index - Cash (DX-Y.NYB) |
| Industrial | BA, CAT, DE, EMR, ETN, GE, HON, LMT, LUV, PNR, | SP500 returns, MACD of stock prices, RSI of stock prices, ETF prices including DIA, EXI, IYJ and VIS |
| Materials | APD, AVY, BLL, DD, ECL, FMC, IFF, IP, NEM, VMC | SP500 returns, MACD of stock prices, RSI of stock prices, gold prices, silver prices, ETF prices including IYM and VAW |

Table 10: Selected stocks and features

## L.2    EXTRA PENALTIES

For real-world experiments, we consider two extra penalty terms for better stability. The first penalty is the model calibration loss discussed in Appendix E.3. Given a trajectory with time interval $\Delta t$, $\tau := \{t_i, s_{t_i}, x_{t_i} \mid i \in [m]\}$, it is defined as

$$-\lambda_1 \min_{C,b} \sum_{i=1}^{m} \left\| \phi(X_{t_{i+1}}) - C\phi(X_{t_i}) - b \right\|_2^2,$$

where $C$ is a matrix and $b$ a vector of proper dimensions. As discussed in Appendix E.3, this penalty encourages a simple representation function $\phi$. The second penalty is the negative sample variance of the terminal wealth, with the parameter $\lambda_2$ determining its strength. The intuition of this penalty is to further penalize the instability of the algorithm performance. The second penalty is implemented for all the competing methods except MMMC for a fair comparison.

### L.3 TRAIN-VALIDATION-TEST SPLIT BY SLIDING WINDOW

We detail the sliding window method for train-validation-test split for real-world portfolio optimization experiments (Figure 7). In financial markets, the dynamics under asset prices and factors vary over time, leading us to construct a sliding window on the dataset for training, validation and testing. Specifically, given a dataset of asset prices and observed features, we construct several windows of observations of equal length. We divide each window into three contiguous periods, the first used for training, the second for validation, and the third for testing. We refer to the length (in days) of the training period as the training size (same for validation and test periods).

After constructing one window, we move the start time point by a fixed number of days (the window gap), and construct the second window. A given method is trained on the training set of each window separately, and then validated and tested on the corresponding validation and test sets. The final validation and test performances are calculated by averaging over each window. The experimental setup is summarized in Table 11. We report the considered hyperparameter values in Table 12.

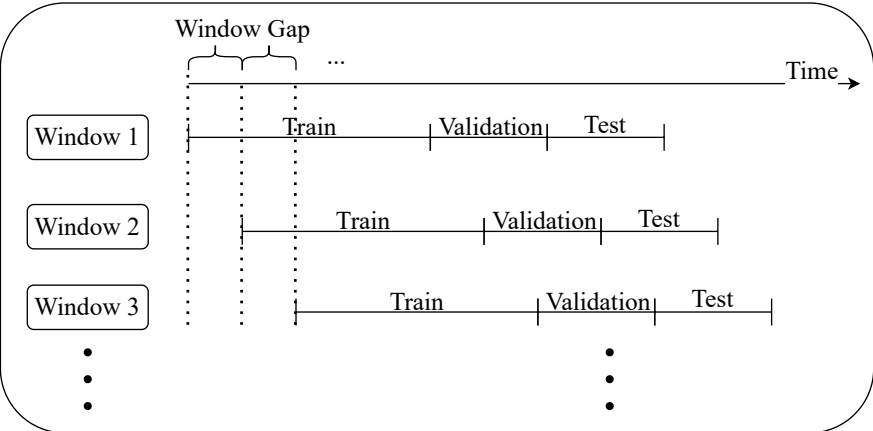

Figure 7: Demonstration of sliding window.

| Experiment Configurations | Values |
|---|---|
| Time Interval $\Delta t$ | 1 (Day) |
| Terminal Time $T$ | 21 |
| Utility Function | $Exponential$ |
| Risk Aversion $\gamma$ | 5 |
| Number of Replications under Each Hyperparameter | 10 |
| Compute Resources | AWS ec2 m5ad.24xlarge |

Table 11: Setup for real-world experiments.

| Hyperparameters | Values |
|:---:|:---:|
| Batch Size | $\{100, 200, 400\}$ |
| $\lambda$ | $\{0.1, 0.5, 0.9\}$ |
| Learning Rate | $\{0.0005, 0.001, 0.01, 0.1\}$ |
| Window Gap | $\{63, 126\}$ |
| Train Size | $\{1260\}$ |
| Validation Size | $\{63\}$ |
| Test Size | $\{63\}$ |

Table 12: Hyperparameters for real-world experiments

### L.4 SENSITIVITY ANALYSIS ON $\lambda$

According to (5), the value of $\lambda$ determines the weight of the FaLPO model calibration. In this section, we conduct sensitivity analysis of FaLPO on $\lambda$. Under the same protocol as the experiments in Section 5.2, we also report FaLPO with different values of $\lambda$ when applied to different sectors. The results are reported in Figure 8. Compared to the case without model calibration ($\lambda = 0$), a small non-zero $\lambda$ provides higher terminal utilities and lower variance. This observation justifies our method of incorporating model calibration into policy learning. Then, when $\lambda$ gets bigger and close to one, the performance of FaLPO decays while the variance also gets smaller.

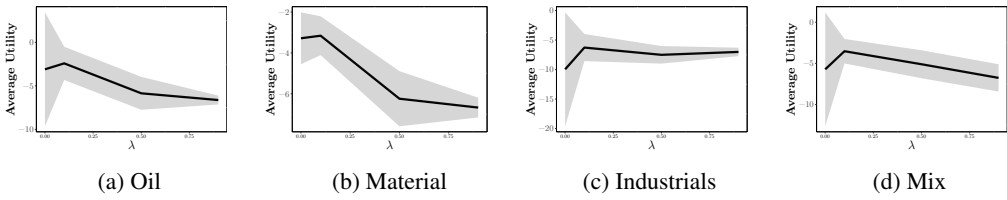

|         (a) Oil         |       (b) Material       |      (c) Industrials      |         (d) Mix         |

Figure 8: Sensitivity analysis for $\lambda$

## M MORE INFORMATION FOR COMPETING METHODS

Here we provide more information for the implemented competing methods. First of all, we focus on policy learning methods, without studying other performance improving techniques like data augmentation or feature engineering. (See Appendix A.2 for a review of such methods.) Such techniques can be easily applied to FaLPO. Further, for a thorough comparison, we summarize the existing policy-learning methods for portfolio optimization with the following four representatives. Note that, all the following methods take asset price data and features as the input for a fair comparison.

- DDPG is implemented with the gradient estimation detailed in Appendix C and also discussed in Nan et al. (2022); Xiong et al. (2018); Jiang et al. (2017). This design makes sure that DDPG can leverage offline data without exploration.

- SLAC (Lee et al., 2020) learns a representation of factors jointly with policy learning. But in this process, no parametric models are used.

- RichID (Mhammedi et al., 2020) falls into the category of model-based policy learning like Yu et al. (2019). It first learns the representation of factors and then conduct policy learning. In this process, both steps take advantage of a parametric model. For better performance in portfolio optimization, we pick Kim-Omberg model as the used model, instead of the LQR model original proposed with this method.

- CT-MB-RL is a policy gradient method optimizing the performance objective using the policy functional form derived from continuous-time models, but without factor representation learning.

We also implement MMMC as a representative of continuous-time finance methods. More complicated and advanced continuous-time finance methods are hard to implement for two reasons. First, to implement such methods, we need to estimate all the parameters of a multivariate SDE (like $\sigma$, $v$, $\mu$ and $\omega$ in Section 3.3). It is challenging since the derivation of likelihood requires solving multivariate stochastic integrals (Ait-Sahalia & Kimmel, 2010), Second, deriving explicit optimal policy functions is also difficult, which involves solving high-dimensional PDEs (like $k_2(t)$ and $k_3(t)$ in Lemma F.2). Further, such pure continuous-time models are expected to underperform, since they assume that the data exactly follow a parametric SDE and tend to underfit. This can also be seen by our comparison with CT-MB-RL (Section 5), which is a model-based RL method by relying on a Kim-Omberg model. As a result, we focus our empirical comparison to more competitive RL methods.

Note that FaLPO circumvents the two aforementioned challenges. First, our model calibration does not aim to fit all the parameters in an SDE, but only those related to learning $\theta_\phi$ and $\theta_\pi$. That is why our model calibration loss in Section 3.3 has such an easy-to-calculate form with the parameter $\theta_S$ as a simple vector. Second, FaLPO does not need a fully derived closed-form solution for the optimal policy. Like in Section 3.3, we use neural networks to parameterize $K(t)$ and $\phi()$, instead of fully deriving them like continuous-time finance methods. Being able to bridge this gap between continuous-time finance models and high multidimensional stock trading problems is one of our contributions.

## N    EXPERIMENTS WITH TRANSACTION COSTS

In this section, we consider the case with transaction costs. Usually, the cost of borrowing a stock to short can vary but typically ranges from $0.3\%$ to $3\%$ per year. Therefore, we take $1\%$ annual transaction cost for short selling an asset. (The fees are applied daily.) Under this setting, we replicate our real-world experiments for the oil sector, using the same protocol. After the tuning procedure in Appendix L.3, the achieved results are reported in Table 13. It should be noticed that the results are consistent with those without transaction costs.

| Methods | Average Utility |
|:---:|:---:|
| **FaLPO** | $-2.25 \pm 1.649$ |
| DDPG | $-6.795 \pm 0.8247$ |
| SLAC | $-7.115 \pm 0.8872$ |
| RichID | $-6.365 \pm 0.5989$ |
| CT-MB-RL | $-5.57 \pm 5.036$ |

Table 13: Average terminal utility in oil sector with transaction costs

## O    EXPERIMENTS WITH DIFFERENT INITIAL WEALTH

We vary the initial wealth in $\{3000, 5000, 8000, 1000\}$ for portfolio optimization using stocks in the oil sector following the same protocol as the experiments in Section 5.2. The results are summarized in Table 14. Specifically, FaLPO achieves superior performance to the competing methods with different initial wealth. Also, it should be noticed that all the methods achieve higher terminal utility given more initial wealth.

| Initial Wealth | 3000 | 5000 | 8000 | 10000 |
|:---:|:---:|:---:|:---:|:---:|
| **FaLPO** | $-21.08 \pm 16.775$ | $-2.4 \pm 1.9$ | $-0.243 \pm 0.1209$ | $-0.03595 \pm 0.008896$ |
| DDPG | $-909.5 \pm 3443.846$ | $-6.6 \pm 1.2$ | $-0.34665 \pm 0.01809$ | $-0.046755 \pm 0.00444$ |
| SLAC | $-11865 \pm 47260.664$ | $-6.8 \pm 0.2$ | $0.35465 \pm 0.07427$ | $-0.04558 \pm 0.00142$ |
| RichID | $-45.51 \pm 8.368$ | $-6.5 \pm 0.1$ | $-0.33125 \pm 0.009791$ | $-0.045365 \pm 0.0001446$ |
| CT-MB-RL | $-28.715 \pm 18.303$ | $-4.2 \pm 6.2$ | $-0.30995 \pm 0.1441$ | $-0.043055 \pm 0.002061$ |

Table 14: Average terminal utility in oil sector with different initial wealth

