# OpenReview forum: "Factor Learning Portfolio Optimization Informed by Continuous-Time Finance Models"
_ICLR.cc/2023/Conference — Submitted to ICLR 2023_

### Official Review · Reviewer_gA89 · 2022-10-24

**Confidence:** 3
**Correctness:** 3
**Technical Novelty And Significance:** 2
**Empirical Novelty And Significance:** 2
**Recommendation:** 6

**Clarity, Quality, Novelty And Reproducibility:**

The proofs and analysis on the theoretical results are provided and complete to my understanding. However, I was not able to find any implementation code for the experimental results.


Typos:
Page 3 paragraph 3: “finitely” frequently


**Strength And Weaknesses:**

Strength:
- Overall the paper is well-written and the structure is fairly clear
- The proposed method appears to be a natural solution that combines the classic continuous-time stochastic factor model with deep reinforcement learning. By design it takes advantage of the continuous-time model which models the dynamics explicitly, and the advantage of the neural networks which learn the complex correlation between the raw features and the useful factors.
- Both convergence guarantees on the theoretical analysis side, and the comparisons with other baselines are provided.

Weakness:
- I found the theoretical analysis section written in a bit confusing way where the exact assumptions are all referred to the sections in the appendix. For example, in the theorem statements, the needed assumptions are delayed until several places in the appendix. It is not clear to me the precise assumptions that are needed.
- In terms of the baselines used, the selected baselines are a bit on the weak side. The authors compared to the baselines that either uses a vanilla stochastic factor model (MMMC), or those use general deep RL only (not designed in particular for stock trading). The authors did not compare with other existing works which also incorporate RL techniques for portfolio optimization or trading, such as the related works mentioned in Appendix A.2. It is unclear if the proposed method is indeed outperforming the state-of-art approaches.

Other questions:
 - I had some confusion when interpreting the numbers in Table 3. It is said that the experiments were conducted with a rolling window, using end-of-day data from 2006 to 2022. Are the numbers in Table 3 averaged values across the entire 2006-2022 period?
- the results are reported in terms of averaged terminal utility. What about the results in terms of average daily/ monthly return?
- How does the proposed method’s performance change in terms of the size of the initial portfolio?

**Summary Of The Paper:**

This paper studies the portfolio optimization problem with factor learning, where the authors use deep reinforcement learning models to learn the factors, and combined that with the classic continuous time finance factor model. The deep RL component is based on the deep deterministic policy learning (DDPG) model from the literature; the stochastic factor model is applied with the learnt factors.

The authors provided theoretical analysis for the proposed framework (asymptotic convergence and finite-sample convergence rate), and conducted comparative experiments with several other baselines using synthetic and real-world stock data.

**Summary Of The Review:**

Overall I think this is a well-written paper which provided a natural solution to the portfolio optimization problem. However I had some concern on the experimental comparison side where some stronger baselines might be missing. I was also not able to find implementation details.

---

### Official Review · Reviewer_TKCJ · 2022-10-24

**Confidence:** 3
**Correctness:** 3
**Technical Novelty And Significance:** 4
**Empirical Novelty And Significance:** 4
**Recommendation:** 3

**Clarity, Quality, Novelty And Reproducibility:**

Overall, the contents of the paper are clearly written and easy to read as it is self-containing.
But the contents balancing needs to be improved.
There are strong technical novelty and originality as it tackles core problems of financial modeling.

**Strength And Weaknesses:**

The overall idea is strong and makes sense, as there are many well-written papers in ML fields incorporating the flexible modeling power of ML models and domain knowledge. It is true that the ML models are prone to fall into overfitting and stochastic models underfitting. And it is a technically challenging procedure to bridge the discrete-time RL models and continuous-time stochastic models. Policy functional form is quite novel as it bridges the discrete and continuous model.

Overall, the paper is easy to read, but the background contents are too much and could be simplified and adequately referenced. It is kind to provide details of finance fields (as it is self-contained), but the amount of appendix is too much.

Experiments seem to be improved. As the Kim-Omberg model could not model every detail(properties) of real-world data, the synthetic experiments seem to be not enough to show the superiority of the proposed method. Also, as the RL models improved over time, the stochastic models also improved over time, but only MMMC (written in 1969) are compared. The FaLPO may capture (cross-sectional) stock selection performance or market timing, but those components are not systematically compared as in finance papers.

**Summary Of The Paper:**

A portfolio optimization method with a stochastic factor model combined with reinforcement learning is proposed. Financial time series have a low signal-to-noise ratio, which is easy to fall into overfitting. The stochastic factor model is robust to such noise as its model noisy terms but it is possible to oversimplify the complex relationships between factors. Meanwhile, the RL approaches try to model every detail and easy to falls into overfitting. FaLPO is proposed to interpolate the RL and stochastic processes to take benefit of the robust modeling power of the stochastic factor model and the flexible modeling power of the RL approach.

**Summary Of The Review:**

Overall, it is technically a novel paper, but the balance of the contents needs to be improved. As many papers in finance literature consume large portion of the robustness checking procedure, experimental results need to be improved.

---

### Official Review · Reviewer_oc9r · 2022-10-25

**Confidence:** 4
**Correctness:** 3
**Technical Novelty And Significance:** 2
**Empirical Novelty And Significance:** 2
**Recommendation:** 3

**Clarity, Quality, Novelty And Reproducibility:**

The work is clear and of the fair quality, but not reproducible because of the source of data and some technical details.

**Strength And Weaknesses:**

* Strength

(i) The work is well written and motivated, compared to a policy network that observes pricing data only, the proposed method involves factors.

(ii) The factors are not manually defined, instead they are learned, within the unified objective that considers both the evolving of factor and asset (through an SDE) and the RL-based loss (through DDPG).

(iii) The comparison against several baselines showed some improvements.

* Weakness

(i) I think the experiment design has a major flaw. The key difference here is that the proposed method can access some external data -- factors, either in raw or derived. Therefore, to make a fair comparison, the authors should keep RL part (DDPG) the same, and try to feed (i) no factors (ii) "off-the-shelf" factors, e.g., Fama-French-Five (iii) "off-line" learned factors, e.g., by fitting SDE or training NN-based model separately. By doing so, we can identify the improvement is indeed credited to the joint learning of factors.

(ii) Some details on backtesting (e.g., transaction cost, as short is allowed -- the cost of borrowing assets, etc.) are missing, and it's a concern why Table 3 (Mix) got some extreme values like 10^8.

**Summary Of The Paper:**

This work proposes Factor Learning Portfolio Optimization (FaLPO), which combines tools from both machine learning and continuous-time finance. It has three learnable parts for: (i) learning factors ($\theta_\phi$) (ii) learning trading policy ($\theta_\pi$) (iii) calibrating SDE ($\theta_S$). The role of factor learning part (i) has an influence on both the objective of (ii) and (iii), therefore these three parts can be jointly optimised. On both synthetic and real-world portfolio optimization tasks, FaLPO outperforms some existing methods.

**Summary Of The Review:**

Overall, I think this paper address an important (and popular) problem in finance, but the current experiment design can not deliver a convincing result.

---

### Decision · Program_Chairs · 2023-01-20

**Decision:**

Reject

**Justification For Why Not Higher Score:**

There are major concerns that have to be addressed before the paper can be accepted. The authors tried to address those during the rebuttal period, but the majority of the reviewers did not find their concerns addressed.

**Justification For Why Not Lower Score:**

N/A

**Metareview: Summary, Strengths And Weaknesses:**

Based on the reviews, it seems that the paper has several strengths, including its well-written motivation, the fact that it combines elements of both reinforcement learning and continuous-time finance methods, and the convergence and performance guarantees provided. However, the reviewers also identified a number of weaknesses in the paper. In particular:

- The experimental design has a flaw because the proposed method can access external data (the factors), and a more fair comparison would be to feed the same RL part (DDPG) with different types of factors (no factors, off-the-shelf factors, or off-line learned factors).

- The experimental results need to be improved, and the FaLPO may capture cross-sectional stock selection performance or market timing, but these components are not systematically compared in the paper.

- The theoretical analysis section is written in a confusing way, with the needed assumptions delayed until several places in the appendix, and the selected baselines are a bit weak and do not include some recent and relevant works.

Overall, the reviewers have identified some areas for improvement in the paper, such as the experiment design and the clarity of the theoretical analysis. It is helpful for the authors to address these issues and revise the paper accordingly before resubmitting it for publication.

**Summary Of Ac-Reviewer Meeting:**

N/A